# Two (narrow) heads are better than (an arbitrarily wide) one[*]

**Amanuel Tesfaye, Zeno Kujawa, Rajmohan Rajaraman & Ravi Sundaram**
Khoury College of Computer Sciences
Northeastern University

## Abstract

In this paper, we establish a dimension- and precision-independent impossibility result for a simplified transformer model. Due to their size, a comprehensive understanding of the internal operations of frontier large language models (LLMs) is beyond the reach of current methods, but research into small and interpretable models has proven successful. We study the representational limits of attention, the core of transformer models, through the lens of the Endpoint Selection Problem (ESP), a simple yet expressive learning task defined over arcs of a directed graph.

Our main theoretical results are twofold: (i) 1-head, 1-layer, attention-only transformers cannot solve ESP on any graph containing a cycle, even with unbounded dimension and precision; but, all DAGs (Directed Acyclic Graph) are solvable with zero error (ii) in contrast, a 2-head, 1-layer, attention-only transformer can solve ESP on arbitrary directed graphs with constant embedding dimension and logarithmic precision. Prior lower bounds (Peng et al., 2024; Sanford et al., 2024b) were conditional on bounds on dimension and precision. Through a transformation, we extend our impossibility result from ESP to the much studied 2-hop induction head problem. Further, we uncover a surprising connection to NP-completeness by showing that the optimal error of the 1-head transformer is exactly related to the size of MAS (Maximum Acyclic Subgraph) and hence inapproximable.

Finally, we validate our theory with experiments and observe that gradient-based optimization can reliably find 1-head solutions for DAGs and 2-head solutions for arbitrary graphs with cycles, whereas 1-head models struggle to reach the optimal solution in graphs with cycles. We believe that our techniques are of independent interest and have the potential to establish a new fine-grained hierarchy of transformer architectures, each with greater problem-solving power than the last.

## 1 Introduction

The transformer architecture (Vaswani et al., 2017) revolutionized artificial intelligence and made possible the astonishing performance of large language models (LLMs). These systems exhibit emergent abilities, reasoning, planning, even apparent world knowledge, that seem disproportionate to the simplicity of their next-token prediction objective. However, despite this success, our understanding of why transformers work remains shallow. Fine-grained analyses of LLMs are far beyond the reach of current techniques. Their sheer scale, training complexity, and data dependence make rigorous proofs almost impossible. To make progress, researchers have turned to simplified transformer models: one or two layers, few attention heads, and limited embedding dimensions, trading emergent phenomena for the possibility of mathematical analysis. Previous work has revealed surprising capabilities of such small transformers in domains such as path-finding (Wang et al., 2024a), learning causal structures (Nichani et al., 2024), and compositional reasoning (Wang et al., 2025). However, unconditional boundaries, results that cleanly separate what a given architecture can and

---

[*]Partially supported by NSF CCF-2335187.

Emails: {tesfaye.am, kujawa.z, r.rajaraman, r.sundaram}@northeastern.edu

cannot do, remain rare. Establishing such hierarchies is a necessary stepping stone toward a principled understanding of how the emergent properties of full-scale LLMs arise and how we can further enhance them.

In this paper, we propose the Endpoint Selection Problem (ESP) as a natural test case. ESP requires selecting an endpoint of an arc in a directed graph. This deceptively simple task is closely tied to graph traversal and selection primitives, which underlie many higher-level reasoning problems, from path-finding to decision making in structured domains. ESP can be generalized in multiple directions: hypergraphs, multi-step traversal, or probabilistic endpoint selection. Thus, understanding transformer performance on ESP is both intrinsically interesting and practically useful as a foundation for analyzing broader classes of algorithmic reasoning tasks.

Our central result establishes an unconditional impossibility: no 1-head, 1-layer, attention-only transformer can solve ESP on any directed graph with cycles. This barrier is independent of precision or embedding dimension (also referred to as *width* in the literature) in contrast to prior results that required stronger assumptions. At the same time, we demonstrate that a modest extension, two heads in a single layer, suffices to solve ESP on all directed graphs, with constant embedding dimension and precision logarithmic in the number of vertices of the graph. Our proof techniques are not limited to ESP; rather, they provide a generalizable framework for analyzing transformer hierarchies. The typical transformer architecture consists of attention layers alternating with FFN (Feed-Forward Network) layers. Since a FFN with a single layer is already a universal approximator (Hornik et al., 1989), in this paper we focus on attention-only transformers a la (Nichani et al., 2024). Recent theoretical work suggests that a single attention layer with an exponential number of heads is a universal sequence-to-sequence approximator (Hu et al., 2025). By charting the landscape of what successively more expressive transformer architectures can and cannot do, we aim to reveal the structure behind the 'magical' emergent properties of LLMs.

## 1.1 SUMMARY OF OUR RESULTS

**Endpoint Selection Problem (ESP).** We define ESP to be the task of correctly predicting the indicated endpoint (head or tail) of an arbitrary arc in a directed graph. The input is an ordered pair (representing the arc), a selector token, and a fixed query token; the output should be the member of the ordered pair specified by the selector token. The input distribution is assumed to be uniform over all arcs and selectors, i.e., uniform over a space of size $2m$, given a directed graph with $m$ arcs. A model is said to solve ESP perfectly, or with zero error, if it always selects the correct endpoint. We set the temperature $\approx 0$ (temperature is the hyperparameter that controls randomness in the output distribution (Hinton et al., 2015)) so that the model is deterministic.

- **1-head transformers**. No model can solve ESP perfectly for any graphs with cycles, even if embedding dimension and precision are unbounded (Theorem 2).
- For any DAG with $n$ vertices, there exists a 1-head model with embedding dimension $O(n)$ that can solve ESP with zero error (Theorem 1).
- No model can solve the well-studied 2-hop induction head problem (Olsson et al., 2022; Sanford et al., 2024c); our proof exploits a transformation from ESP to 2-hop induction head (Corollary 1).
- The optimal 1-head model's error is exactly $\frac{1}{2} - \frac{|\text{MAS}|}{2m}$, where MAS is the Maximum Acyclic Subgraph, an NP-complete problem (Theorem 1 and Corollary 2). It is NP-complete to even approximate the minimum error 1-head model for an arbitrary directed graph (Theorem 5).
- **2-head transformers**. We prove that a 2-head 1-layer attention-only transformer can solve ESP without error for any $n$-vertex directed graph, using $O(n)$ dimension and $O(1)$ precision (Theorem 3) or using $O(1)$ embedding dimension and $O(\log n)$ precision (Corollary 2).
- **Empirical analysis**. Experiments corroborate our theoretical results – gradient-based optimization can reliably find 1-head solutions for ESP on DAGs and 2-head solutions for ESP on arbitrary graphs with cycles, whereas 1-head models struggle to reach the optimal solution in cyclic graphs.

Our ESP formulation is arguably the simplest problem yielding the impossibility and intractability results with attention-only models that we have derived. It enables us to extend our lower bounds to related well-studied problems. Furthermore, the graph-theoretic formulation establishes a deeper connection between the representational capacity of a transformer for a given instance to the underlying graph's structural properties.

## 1.2 RELATED WORK

Our work contributes to the understanding of the capabilities of small transformer models. Several positive and negative results on transformer capabilities have been established in this field.

**Transformer capabilities on graphs**. In Wang et al. (2024a), it is demonstrated that constant-depth transformer models can be trained to find paths in a directed acyclic graph (DAG). The authors prove their claim by proposing a set of weights capable of achieving this task and showing that this set of weights can be found by using gradient descent. A classification of 9 graph problems grouped by the model scales sufficient to solve them can be found in Sanford et al. (2024a). In Nichani et al. (2024), it is shown that a two-layer attention-only transformer model can learn latent causal structure in sequences. Further, they show that the transformer learns an induction head (Olsson et al., 2022) when sequences are generated from an in-context Markov chain.

**Comparison to Index Lookup**. Bhattamishra et al. (2024) analyze the Index Lookup problem, where a model must return the token at a specified position in a sequence using the index token as the query, and show that a 1-layer, 1-head transformer augmented with a small FFN and logarithmic width can implement this task. Despite the similarity, the Index problem is different from ESP, since in ESP the query token prevents the model from relying on **direct key–query alignment** between the selector token and the corresponding sequence position (or arc endpoint). Indeed, our impossibility argument for ESP provides a sharp separation from the logarithmic-width, logarithmic-precision model for Index in Bhattamishra et al. (2024). A second significant difference between the two formulations is that our ESP problem framework is defined over graphs and captures the limitations of one-head transformers and capability of two-head transformers to resolve the problem defined over different graph classes. In particular, our inapproximability result (Theorem 5) indicates that even finding a 1-head model that approximates accuracy to within a 1.3606 factor is intractable. This enables us to identify limitations in both the representation of the model as well as the intractability of finding weights in the model that will lead to a desired level of accuracy.

**Lower bound results**. A lower bound for function composition is formally proved in Peng et al. (2024); using communication complexity arguments, they show that for attention-only 1-layer transformer models with embedding dimension $d$, input domain size $|A| = |B| = |C| = n$, with numbers represented with $p$ bits, and $H$ heads, function composition is impossible if $H(d + 1)p < n \lg n$. Chen et al. (2024) extends this result to standard multi-layer transformers by showing that an $L$-layer transformer with sequence length $n$ needs model dimension $n^{\Omega(1)}$ to compose $L$ functions.

Sanford, Hsu, and Telgarsky prove several complexity results for transformers in Sanford et al. (2023; 2024a) and Sanford et al. (2024c). They show that the 1-hop induction head task (which is the task of returning, for two sequences of tokens $(\sigma_1, \ldots, \sigma_n)$ and $(\tau_1, \ldots, \tau_n)$, where $\tau_i$ is the input token that follows the rightmost occurrence of the token equal to $\sigma_i$ before position $i$) cannot be solved by a 1-layer attention-only transformer unless $mph = \Omega(n)$, where $m$ is the embedding dimension, $p$ is the precision, and $h$ is the number of attention heads. Since it is known from previous work that a 2-layer attention-only transformer with $h = O(1)$, $m = O(1)$, $p = O(\log(n))$ can solve this task, the size lower bound for 1-layer attention-only transformers is exponentially larger than for 2-layer attention-only transformers (Sanford et al., 2024b; Bietti et al., 2023). In Sanford et al. (2023), the tasks Match2 and Match3 are introduced, in which for a sequence $X = (x_1, \ldots, x_n) \in [M]^N$ the output is a vector $V$ where $V_i = 1$ iff there is a pair of elements in $X$ such that $x_i + x_j = 0$ mod $M$ (or a triplet in the case of Match3). They then show that Match2 can be solved by a 1-layer 1-head transformer, while a single-layer multihead transformer is not sufficient for Match3. It is shown in Kozachinskiy et al. (2025) that even with infinite precision, a single-layer transformer with size $n^{O(1)}$ can solve neither Match3 nor the composition task given in Peng et al. (2024).

**Proofs of transformer advantages**. In Sanford et al. (2023), the authors propose an averaging task that demonstrates the performance gained by increasing embedding dimension as well as an efficiency improvement of an attention-only 1-layer transformer over recurrent and fully connected neural networks. This result is expanded upon in Wang et al. (2024b), where a simpler version of this task is used to show that it is efficiently learnable with a convergence guarantee for a 1-layer attention-only transformer, while a fully-connected neural network must have exponentially more neurons in its first layer than the minimum width required for the transformer.

**Circuit complexity results**. Another set of complexity-theoretic results for Transformers can be found in Merrill & Sabharwal (2023; 2024a;b). In Merrill & Sabharwal (2023), it is shown that transformers with constant depth, context length $n$, and precision $O(\log n)$ can be simulated by uniform constant-depth threshold circuits, thus proving they cannot solve problems beyond uniform $TC^0$ such as graph connectivity. The same is shown to be true for transformers using average-hard attention in Merrill et al. (2022), which is extended to uniform TC$^0$ in Strobl (2023). Furthermore, in Chiang (2025), it is shown that transformers with either attention score computation method are in DLOGTIME-uniform TC$^0$, with softmax attention assuming $O(poly(n))$-precision. It was shown in Merrill & Sabharwal (2024a) that log-depth transformers can solve graph connectivity, improving our understanding of the performance improvements attainable with increased depth. They also show that for a fixed-depth transformer, the hidden dimension must grow superpolynomially with input length in order for the transformer to solve regular language recognition and graph connectivity. Furthermore, they prove that a transformer with $O(\log n)$ chain-of-thought steps cannot solve any problem outside of $TC^0$. A function composition task is discussed in Wang et al. (2025), where the authors show that a complex compositional task called *k-fold composition* can be learned by a transformer with $O(\log k)$ layers. In Chen et al. (2025), it is shown that a constant-depth transformer of $poly(n)$-size with $poly(n)$-precision using rotary position embeddings (RoPE) can be simulated by a DLOGTIME-uniform TC$^0$ circuit family unless TC$^0$ = NC$^1$. Other circuit complexity results are presented in Cao et al. (2025); Hahn (2020); Strobl et al. (2024).

**Empirical evaluation of multi-head attention**. In Michel et al. (2019), the authors show that for some tasks, transformer models do not exploit the flexibility provided by multi-head attention and do not suffer from significant performance degradation when pruned from many heads to fewer heads per layer. Several other work exists in this area, such as Liu et al. (2021) and Voita et al. (2019).

## 2 PROBLEM SETUP AND TRANSFORMER MODEL

To analyze the representational power of attention heads, we introduce the **Endpoint Selection Problem (ESP)**, a supervised learning task defined over arcs of a directed graph $\mathcal{G} = (V, E)$. Its vocabulary set $\mathcal{S}$ consists of:

$$\mathcal{S} = \underbrace{\{v_1, \ldots, v_n\}}_{\text{vertex tokens } \mathcal{V}} \quad \cup \quad \underbrace{\{1, 2\}}_{\text{indicator tokens } \mathcal{I}} \quad \cup \quad \underbrace{\{\#\}}_{\text{query token}} \quad .$$

Given an arc $(u, v) \in E$, an indicator $i \in \mathcal{I}$, and the query $\#$, the input sequence is $(u, v, i, \#)$, and the model must output $u$ if $i = 1$ and $v$ if $i = 2$.

Fixing the query token restricts the model, effectively transforming ESP to a 2-hop induction head task (as we show in Appendix A). This task, studied in previous work (Sanford et al., 2024c), is a natural extension of the original induction head problem analyzed in Olsson et al. (2022). Its more general form, the $k$-hop induction head, is closely related to other sequential reasoning problems, such as pointer chasing (Sanford et al., 2024c; Peng et al., 2024) and $k$-fold composition (Wang et al., 2025).

**Transformer model**. For brevity, we follow notation similar to Nichani et al. (2024). Let the input sequence be $S_{1:T} := (s_1, \ldots, s_T) \in \mathcal{S}^T$, where $\mathcal{S}$ is the vocabulary. The sequence is embedded as $\boldsymbol{X} := \text{embed}(S_{1:T}; \boldsymbol{E}, \boldsymbol{P})$, with:

$$\boldsymbol{X}_i := \boldsymbol{E}_{s_i} + \boldsymbol{P}_i \quad \text{for } i = 1, \ldots, T, \quad \text{where } \boldsymbol{X}, \boldsymbol{P} \in \mathbb{R}^{T \times d} \text{ and } \boldsymbol{E} \in \mathbb{R}^{|\mathcal{S}| \times d}.$$

where $\boldsymbol{E}$ and $\boldsymbol{P}$ are the token and positional embedding matrices, respectively. A single layer attention-only transformer consists of a Multi-Head Attention (MHA) mechanism which computes the weighted sum of value vectors across $k$ heads:

$$\text{MHA}(\boldsymbol{X}) := \sum_{j=1}^{k} \text{Softmax}\left(\text{Mask}(\boldsymbol{X} \boldsymbol{A}_j \boldsymbol{X}^\top)\right) \boldsymbol{X} \boldsymbol{V}_j \in \mathbb{R}^{T \times d_{out}},$$

where $\boldsymbol{Q}, \boldsymbol{K}, \boldsymbol{V}$ are the query, key, and value parameter matrices of the attention mechanism and $\boldsymbol{A}_j := \boldsymbol{Q}_j \boldsymbol{K}_j^\top \in \mathbb{R}^{d \times d}$. The output of the MHA block is passed through a final linear layer (parameterized by $\boldsymbol{W}_0$) to produce the output logits over the vocabulary.

$$\boldsymbol{Z} := \text{TF}_\theta(S_{1:T}) := \text{MHA}(\text{embed}(S_{1:T}; \boldsymbol{E}, \boldsymbol{P})) \boldsymbol{W}_0^\top \in \mathbb{R}^{T \times |\mathcal{S}|}$$

The full set of model parameters is $\theta = (\boldsymbol{E}, \boldsymbol{P}, \{\boldsymbol{A}_j\}_{j=1}^k, \{\boldsymbol{V}_j\}_{j=1}^k, \boldsymbol{W}_0)$. The final predicted token, $\hat{s}$, is selected by finding the token with the highest logit score at the final sequence position $T$. We take the limit of the softmax distribution as the temperature approaches zero, which is equivalent to the $\arg\max$ operation: $\hat{s} := \arg\max_{s' \in \mathcal{S}} \boldsymbol{Z}_T[s']$.

## 3 ANALYSIS OF 1-HEAD TRANSFORMERS

### 3.1 1-HEAD MODELS CAN SOLVE ESP OVER DAGS

In this subsection, we prove that a 1-head transformer model can solve the selection problem over DAGs, if the dimension of the embedding space is at least the number of vertices in the DAG. In fact, we establish a more general result by quantifying the least error that can be achieved by a 1-head model on arbitrary directed graphs.

**Theorem 1.** *For any integer $n$ and any directed graph $G$, which has $n$ vertices, $m$ edges, and an acyclic subgraph with $m'$ edges, there exists a 1-head transformer model with embedding dimension $n + 1$ that incurs error at most $1/2 - m'/(2m)$ for ESP on $G$.*

*Proof.* Let $G$ be a directed graph over a set $V$ of $n$ vertices and $m$ edges, and let $H$ be an acyclic subgraph of $G$ with $m'$ edges. Consider the following labeling of the vertices of $G$: vertex $v_i$ denotes the $i$th vertex in an arbitrary topological ordering of $H$; so any edge $(v_i, v_j)$ in $H$ satisfies $i < j$.

Our construction uses both a token embedding and a positional embedding. The embedding dimension is $d = n + 1$. For each token $v_i$, the token embedding is simply the unit vector with 1 in dimension $i$. For tokens 1 and 2, we set the embeddings to be the following vectors, respectively.

$$\alpha \begin{pmatrix} \frac{n}{n} & \cdots & \frac{n-i}{n} & \cdots & \frac{1}{n} & \gamma \end{pmatrix}^T \text{ and } \alpha \begin{pmatrix} \frac{1}{n} & \cdots & \frac{i}{n} & \cdots & \frac{n}{n} & 0 \end{pmatrix}^T,$$

for parameters $\alpha$ and $\gamma$ which we will set shortly. For the query token #, we set the embedding to be the vector $\begin{pmatrix} 1 & \cdots & 1 \end{pmatrix}^T$. The positional embedding for position 1 is $\begin{pmatrix} 0 & \cdots & 0 & \delta \end{pmatrix}^T$, and zero for all other positions, where $\delta$ will be specified later in the proof.

The attention matrix $\boldsymbol{A}$ is set to $\boldsymbol{I}$. We now conduct the output analysis for an input of the form $(v_i, v_j, s, \#)$, where $s \in \{1, 2\}$ represents the selector. We determine the attention weights as follows:

$$\boldsymbol{e}(\#)^T \boldsymbol{A} \boldsymbol{e}(v_i) = 1 + \delta; \ \boldsymbol{e}(\#)^T \boldsymbol{A} \boldsymbol{e}(v_j) = 1; \ \boldsymbol{e}(\#)^T \boldsymbol{A} \boldsymbol{e}(\#) = n + 1;$$
$$\boldsymbol{e}(\#)^T \boldsymbol{A} \boldsymbol{e}(1) = \alpha(n+1)/2 + \alpha\gamma; \ \boldsymbol{e}(\#)^T \boldsymbol{A} \boldsymbol{e}(2) = \alpha(n+1)/2.$$

Here, $\boldsymbol{e}(\cdot)$ simply denotes the embedding vector for each token. Following the softmax calculation, these attention weights lead to the convex coefficients for $v_i$, $v_j$, selector $s$, and # as

$$w_{v_i} = \frac{e^{1+\delta}}{\sigma_s}, w_{v_j} = \frac{e}{\sigma_s}, w_1 = \frac{e^{\alpha(n+1)/2+\alpha\gamma}}{\sigma_s}, w_2 = \frac{e^{\alpha(n+1)/2}}{\sigma_s}, w_\# = \frac{e^{n+1}}{\sigma_s}, \text{ where}$$

$$\sigma_1 = e + e^\delta + e^{\alpha(n+1)/2+\alpha\gamma} + e^{(n+1)} \text{ and } \sigma_2 = e + e^\delta + e^{\alpha(n+1)/2} + e^{(n+1)}.$$

The final context vector $y$ can then be written as $w_{v_i}\boldsymbol{e}(v_i) + w_{v_j}\boldsymbol{e}(v_j) + w_s\boldsymbol{e}(s) + w_\#\boldsymbol{e}(\#)$. Let $y_\ell$ denote the $\ell$-th component of the vector $\boldsymbol{y}$. Then, we can expand the vector $y$ as

$$y_\ell = \begin{cases} \frac{1}{\sigma_1}\left(e^{1+\delta} + \alpha e^{\alpha(n+1)/2+\alpha\gamma}(\frac{n-i+1}{n}) + e^{n+1}\right) & \ell = i \text{ and } s = 1, \\ \frac{1}{\sigma_2}\left(e^{1+\delta} + \alpha e^{\alpha(n+1)/2}(\frac{i}{n}) + e^{n+1}\right) & \ell = i \text{ and } s = 2, \\ \frac{1}{\sigma_1}\left(e + \alpha e^{\alpha(n+1)/2+\alpha\gamma}(\frac{n-j+1}{n}) + e^{n+1}\right) & \ell = j \text{ and } s = 1, \\ \frac{1}{\sigma_2}\left(e + \alpha e^{\alpha(n+1)/2}(\frac{j}{n}) + e^{n+1}\right) & \ell = j \text{ and } s = 2, \\ \frac{1}{\sigma_s}\left(\alpha e^{\alpha(n+1)/2}(\frac{n-\ell+1}{n}) + e^{n+1}\right) & \text{otherwise.} \end{cases}$$

We set $\boldsymbol{V}$ and $\boldsymbol{W}_0$ to $\boldsymbol{I}$ (noting that $d_{out} = d$); set $\alpha = 2/(n+1)$, $\delta < \ln(1 + 2/(n^2 + n))$ to obtain

$$e > \alpha e^{\alpha(n+1)/2} \text{ and } \alpha e^{\alpha(n+1)/2} > n\left(e^{1+\delta} - e\right).$$

These ensure that $y_i, y_j > y_\ell$ for $\ell \neq i, j$ and, for the selector $s = 2$, we have $y_j > y_i$ whenever $i < j$. Therefore, the model works correctly for all input instances $(v_i, v_j, 2, \#)$ whenever $i < j$. By choosing $\gamma < \frac{n+1}{2}(\ln(\frac{n+1}{2}) + \ln(e^\delta - 1))$, we satisfy $e^{1+\delta} > e + \alpha e^{\alpha(n+1)/2+\alpha\gamma}$, so that for any input instance $(v_i, v_j, 1, \#)$, we have $y_i > y_j$, ensuring that the model works correctly for this instance for all $i \neq j$. Thus, of the $2m$ input instances, the model incurs an error on $m - m'$ of the instances, leading to an error of $1/2 - m'/(2m)$, completing the proof of the theorem. $\square$

## 3.2 No 1-head model can solve ESP over any graph with cycles

In this section, we establish that no 1-head 1-layer attention-only transformer model can solve the selection problem over any graph with cycles.

**Lemma 1.** *For any vectors $\boldsymbol{x}_a$ and $\boldsymbol{x}_b$, there do not exist vectors $\boldsymbol{x}_1$, $\boldsymbol{x}_2$, $\boldsymbol{x}_{ab}$ and $\boldsymbol{x}_{ba}$ and non-negative reals $r_1$, $r_2$, $r_{ab}$, and $r_{ba}$ satisfying the following four conditions:*

$$(r_1\boldsymbol{x}_1 + r_{ab}\boldsymbol{x}_{ab}) \cdot (\boldsymbol{x}_a - \boldsymbol{x}_b) > 0, (r_2\boldsymbol{x}_2 + r_{ab}\boldsymbol{x}_{ab}) \cdot (\boldsymbol{x}_a - \boldsymbol{x}_b) < 0$$
$$(r_1\boldsymbol{x}_1 + r_{ba}\boldsymbol{x}_{ba}) \cdot (\boldsymbol{x}_a - \boldsymbol{x}_b) < 0, (r_2\boldsymbol{x}_2 + r_{ba}\boldsymbol{x}_{ba}) \cdot (\boldsymbol{x}_a - \boldsymbol{x}_b) > 0$$

*Proof.* Consider the two-dimensional plane spanned by the vectors $\boldsymbol{x}_a$ and $\boldsymbol{x}_b$; we refer to this as the $x$-$y$ plane. Since the four conditions concern dot products with $\boldsymbol{x}_a - \boldsymbol{x}_b$, it is sufficient to consider the projections of $\boldsymbol{x}_1, \boldsymbol{x}_2, \boldsymbol{x}_{ab}, \boldsymbol{x}_{ba}$ on the $x$-$y$ plane so that we can assume that $\boldsymbol{x}_1, \boldsymbol{x}_2, \boldsymbol{x}_{ab}, \boldsymbol{x}_{ba}$ all lie on the $x$-$y$ plane.

We first establish the claim for the case where $\boldsymbol{x}_a$ and $\boldsymbol{x}_b$ are orthogonal to each other, and then extend the argument to the general case. If $\boldsymbol{x}_a$ and $\boldsymbol{x}_b$ are orthogonal to each other, then without loss of generality, let $\boldsymbol{x}_a$ and $\boldsymbol{x}_b$ be along the $x$- and $y$-axes, respectively. Furthermore, we can set $\boldsymbol{x}_a$ and $\boldsymbol{x}_b$ to be unit vectors by scaling the $x$- and $y$-projections of other vectors without changing any of the dot products.

For any $\boldsymbol{v}$, let $x_v$ and $y_v$ be the projections of $\boldsymbol{v}$ on the $x$- and $y$-axes, and let $\Delta_v$ denote $x_v - y_v$. Then, the first condition can be rewritten as $r_1(x_1 - y_1) + r_{ab}(x_{ab} - y_{ab}) > 0$. Thus, all the four conditions can be rewritten as:

$$r_1\Delta_1 + r_{ab}\Delta_{ab} > 0; \quad r_2\Delta_2 + r_{ab}\Delta_{ab} < 0; \quad r_1\Delta_1 + r_{ba}\Delta_{ba} < 0; \quad r_2\Delta_2 + r_{ba}\Delta_{ba} > 0.$$

Adding the first and fourth inequalities and subtracting the second and third inequalities yields $0 > 0$, a contradiction. It thus follows that the four conditions cannot be simultaneously satisfied.

We now consider the case where $\boldsymbol{x}_a$ and $\boldsymbol{x}_b$ are not orthogonal. Note that each of the four conditions is a requirement that the dot product of $\boldsymbol{x}_a - \boldsymbol{x}_b$ with a specific vector, which is independent of $\boldsymbol{x}_a$ and $\boldsymbol{x}_b$, is either positive or negative. For any $\boldsymbol{x}_a$ and $\boldsymbol{x}_b$, we can find orthogonal vectors $\boldsymbol{x}'_a$ and $\boldsymbol{x}'_b$ satisfying $\boldsymbol{x}'_a - \boldsymbol{x}'_b = \boldsymbol{x}_a - \boldsymbol{x}_b$. This reduces the general case to that where $\boldsymbol{x}_a$ and $\boldsymbol{x}_b$ are orthogonal, thus completing the proof of the lemma. $\square$

**Lemma 2.** *For any cycle $C = (V_C, E_C)$, there do not exist vectors $\boldsymbol{x}_1$, $\boldsymbol{x}_2$, a set of vectors $\{\boldsymbol{x}_{uv} : (u, v) \in E_C\}$, a set of vectors $\{\boldsymbol{x}_v : v \in V_C\}$, and reals $r_1, r_2, \{r_{uv} : (u, v) \in E_C\}$ such that for every $(u, v) \in E_C$:*

$$(r_1\boldsymbol{x}_1 + r_{uv}\boldsymbol{x}_{uv}) \cdot (\boldsymbol{x}_u - \boldsymbol{x}_v) > 0 > (r_2\boldsymbol{x}_2 + r_{uv}\boldsymbol{x}_{uv}) \cdot (\boldsymbol{x}_u - \boldsymbol{x}_v).$$

*Proof.* The proof is by induction on the length of the cycle. For the induction base, we consider a cycle of length 2 with two edges $(a, b)$ and $(b, a)$. The non-existence of the vectors and reals satisying the desired conditions follows from Lemma 1.

For the induction step, suppose the claim holds for all cycles of length at most $k$ where $k \geq 2$. Consider a cycle $C$ of length $k + 1$. For the sake of contradiction, suppose there exist vectors and reals satisfying the conditions stated in Lemma 2.

Let $a$, $b$, and $c$ be contiguous vertices on the cycle. Then, we have the following inequalities hold.

$$(r_1\boldsymbol{x}_1 + r_{ab}\boldsymbol{x}_{ab}) \cdot (\boldsymbol{x}_a - \boldsymbol{x}_b) > 0, \quad (r_2\boldsymbol{x}_2 + r_{ab}\boldsymbol{x}_{ab}) \cdot (\boldsymbol{x}_a - \boldsymbol{x}_b) < 0$$
$$(r_1\boldsymbol{x}_1 + r_{bc}\boldsymbol{x}_{bc}) \cdot (\boldsymbol{x}_b - \boldsymbol{x}_c) > 0, \quad (r_2\boldsymbol{x}_2 + r_{bc}\boldsymbol{x}_{bc}) \cdot (\boldsymbol{x}_b - \boldsymbol{x}_c) < 0.$$

Adding the first and third inequalities, and adding the second and fourth inequalities yield:

$$r_1\boldsymbol{x}_1 \cdot (\boldsymbol{x}_a - \boldsymbol{x}_c) + r_{ab}\boldsymbol{x}_{ab} \cdot (\boldsymbol{x}_a - \boldsymbol{x}_b) + r_{bc}\boldsymbol{x}_{bc} \cdot (\boldsymbol{x}_b - \boldsymbol{x}_c) > 0$$
$$r_2\boldsymbol{x}_2 \cdot (\boldsymbol{x}_a - \boldsymbol{x}_c) + r_{ab}\boldsymbol{x}_{ab} \cdot (\boldsymbol{x}_a - \boldsymbol{x}_b) + r_{bc}\boldsymbol{x}_{bc} \cdot (\boldsymbol{x}_b - \boldsymbol{x}_c) < 0.$$

Set $r_{ac}$ and $\boldsymbol{x}_{ac}$ so that $r_{ac}\boldsymbol{x}_{ac} \cdot (\boldsymbol{x}_a - \boldsymbol{x}_c) = r_{ab}\boldsymbol{x}_{ab} \cdot (\boldsymbol{x}_a - \boldsymbol{x}_b) + r_{bc}\boldsymbol{x}_{bc} \cdot (\boldsymbol{x}_b - \boldsymbol{x}_c)$ ensuring

$$(r_1\boldsymbol{x}_1 + r_{ac}\boldsymbol{x}_{ac}) \cdot (\boldsymbol{x}_a - \boldsymbol{x}_c) > 0 > (r_2\boldsymbol{x}_2 + r_{ac}\boldsymbol{x}_{ac}) \cdot (\boldsymbol{x}_a - \boldsymbol{x}_c).$$

We thus have found a solution to the set of inequalities for a cycle of length $k$ that is obtained by replacing edges $(a, b)$ and $(b, c)$ with $(a, c)$. This contradicts the induction hypothesis. $\square$

**Theorem 2.** *For any directed graph $G$ with cycles, there is no 1-head 1-layer attention-only transformer model that can solve the selection problem over $G$, even with unbounded dimension and unbounded precision.*

*Proof.* Suppose there exists a 1-head model that accurately solves the selection problem over a directed graph $G$ with cycles. Let $C$ be an arbitrary cycle in $G$. Then, in particular, the model accurately solves the endpoint selection problem for every directed edge in $C$.

**Setup**. Fix an arbitrary directed edge $(a, b)$ in $C$. Consider the following two input sequences of length $T = 4$: $S_1 = (a_1, b_2, 1_3, \#_4)$ which must output $a$; $S_2 = (a_1, b_2, 2_3, \#_4)$ which must output $b$. Let the pre-softmax score for a token $s$ at position $p$ be denoted as $z_{s_p} = \#\mathbf{A}x_{s_p}^\top$. For the two sequences, the attention weights are given by:

$$\text{Softmax}\big(\#\mathbf{A}\boldsymbol{x}_{a_1}^\top, \#\mathbf{A}\boldsymbol{x}_{b_2}^\top, \#\mathbf{A}\boldsymbol{x}_{1_3}^\top, \#\mathbf{A}\boldsymbol{x}_{\#_4}^\top\big) = \text{Softmax}(z_{a_1}, z_{b_2}, z_{1_3}, z_{\#_4}),$$

$$\text{Softmax}\big(\#\mathbf{A}\boldsymbol{x}_{a_1}^\top, \#\mathbf{A}\boldsymbol{x}_{b_2}^\top, \#\mathbf{A}\boldsymbol{x}_{2_3}^\top, \#\mathbf{A}\boldsymbol{x}_{\#_4}^\top\big) = \text{Softmax}(z_{a_1}, z_{b_2}, z_{2_3}, z_{\#_4}).$$

These weights determine the final output vectors, and a single-head model must learn one matrix $\boldsymbol{A}$ that works for all the $2|C|$ instances corresponding to the edges in $C$.

**Preliminaries**. The final context vector for each sequence, $S_i$, is the weighted sum of the input embeddings, $\boldsymbol{v}_i = w_i \cdot \boldsymbol{X}_i$. The two context vectors can then be written as:

$$\boldsymbol{v}_1 = \left(\frac{e^{z_{a_1}}\boldsymbol{x}_{a_1} + e^{z_{b_2}}\boldsymbol{x}_{b_2} + e^{z_{\#_4}}\boldsymbol{x}_{\#_4}}{Z_1}\right) + \left(\frac{e^{z_{1_3}}\boldsymbol{x}_{1_3}}{Z_1}\right)$$

$$\boldsymbol{v}_2 = \left(\frac{e^{z_{a_1}}\boldsymbol{x}_{a_1} + e^{z_{b_2}}\boldsymbol{x}_{b_2} + e^{z_{\#_4}}\boldsymbol{x}_{\#_4}}{Z_2}\right) + \left(\frac{e^{z_{2_3}}\boldsymbol{x}_{2_3}}{Z_2}\right)$$

where $Z_i = \sum_{j \in S_i} e^{z_j}$ for $i \in \{1, 2\}$. To simplify the above expression let us define vector, $\boldsymbol{x}_{ab}$, which represents the attention-weighted sum over the non-indicator tokens:

$$\boldsymbol{x}_{ab} := \frac{e^{z_{a_1}}\boldsymbol{x}_{a_1} + e^{z_{b_2}}\boldsymbol{x}_{b_2} + e^{z_{\#_4}}\boldsymbol{x}_{\#_4}}{e^{z_{a_1}} + e^{z_{b_2}} + e^{z_{\#_4}}}$$

Using these definitions, we can express each of the final context vectors as a convex combination of the newly defined vectors and the indicator tokens.

$$\boldsymbol{v}_1 = \left(\frac{e^{z_{a_1}} + e^{z_{b_2}} + e^{z_{\#_4}}}{Z_1}\right)\boldsymbol{x}_{ab} + \left(\frac{e^{z_{1_3}}}{Z_1}\right)\boldsymbol{x}_{1_3} := \frac{1}{Z_1}\left(r_{ab}\boldsymbol{x}_{ab} + r_1\boldsymbol{x}_{1_3}\right)$$

$$\boldsymbol{v}_2 = \left(\frac{e^{z_{a_1}} + e^{z_{b_2}} + e^{z_{\#_4}}}{Z_2}\right)\boldsymbol{x}_{ab} + \left(\frac{e^{z_{2_3}}}{Z_2}\right)\boldsymbol{x}_{2_3} := \frac{1}{Z_2}\left(r_{ab}\boldsymbol{x}_{ab} + r_2\boldsymbol{x}_{2_3}\right)$$

**Geometric interpretation**. The selection between tokens $a$ and $b$ depends on which side of the decision boundary $\boldsymbol{v} \cdot \boldsymbol{x}_a = \boldsymbol{v} \cdot \boldsymbol{x}_b$ the context vector $\boldsymbol{v}$ lies. The two resulting regions are the $a$-dominant half-space, $\mathcal{H}_a = \{\boldsymbol{v}|\boldsymbol{v} \cdot \boldsymbol{x}_a > \boldsymbol{v} \cdot \boldsymbol{x}_b\}$, and the $b$-dominant half-space, $\mathcal{H}_b = \{\boldsymbol{v}|\boldsymbol{v} \cdot \boldsymbol{x}_b > \boldsymbol{v} \cdot \boldsymbol{x}_a\}$. The two selection tasks for any edge $(a, b)$ impose the following constraints on the context vectors $\boldsymbol{v}_1$ and $\boldsymbol{v}_2$: $\boldsymbol{v}_1 \in \mathcal{H}_a$ and $\boldsymbol{v}_2 \in \mathcal{H}_b$. Therefore, we get the following inequality constraints for every edge $(a, b) \in C$.

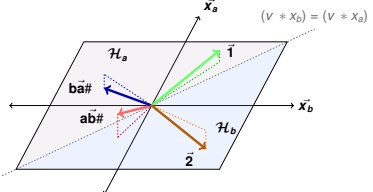

$$\boldsymbol{v}_1 \cdot (\boldsymbol{x}_a - \boldsymbol{x}_b) > 0 \implies (r_{ab}\boldsymbol{x}_{ab} + r_1\boldsymbol{x}_1) \cdot (\boldsymbol{x}_a - \boldsymbol{x}_b) > 0,$$

$$\boldsymbol{v}_2 \cdot (\boldsymbol{x}_a - \boldsymbol{x}_b) < 0 \implies (r_{ab}\boldsymbol{x}_{ab} + r_2\boldsymbol{x}_2) \cdot (\boldsymbol{x}_a - \boldsymbol{x}_b) < 0.$$

By Lemma 2, there do not exist vectors $\{\boldsymbol{x}_a : a \in C\}$, $\{\boldsymbol{x}_{ab} : (a, b) \in C\}$, $\boldsymbol{x}_1$, and $\boldsymbol{x}_2$ and non-negative reals $\{r_{ab} : (a, b) \in C\}$, $r_1$, $r_2$ that satisfy the above set of $2|C|$ constraints. $\square$

Figure 1: 2–D decision plane.

**Corollary 1.** *There is no 1-head 1-layer attention-only transformer model that can solve the 2-hop induction head problem, even with unbounded dimension and unbounded precision.*

*Proof.* Due to space constraints, we provide a brief proof sketch and defer all details, including a formal definition of 2-hop induction head and the full proof, to Appendix A. We transform an instance of ESP to a 2-hop induction head instance as follows: $(a_1, b_2, i_3, \#_4) \Rightarrow (1_1, a_2, 2_3, b_4, \#_5, i_6, \#_7)$. In particular, note that in the 2-hop induction head instances created, the tokens in positions 1, 3, and 5 are always the same. Hence, any transformer circuit makes the next-token prediction based on the tokens in positions 2 and 4, position 6 (which contains token 1 or token 2) and the preceding token (which is $\#$). This is identical to the transformer model that is solving an ESP instance.

We formalize this idea, following the proof technique of Theorem 2. Consider the following two input sequences $(1_1, a_2, 2_3, b_4, \#_5, 1_6, \#_7)$ which must output $a$ and $(1_1, a_2, 2_3, b_4, \#_5, 2_6, \#_7)$ which must output $b$. We derive the attention weights and define context vectors $\boldsymbol{v}_1$ and $\boldsymbol{v}_2$ corresponding to the two sequences and vector $\boldsymbol{x}_{ab}$, the attention-weighted sum over the non-indicator tokens. This enables us to express each of the final context vectors as a convex combination of $\boldsymbol{x}_{ab}$ and the indicator token vectors. (See Appendix A for the derivations.)

$$\boldsymbol{v}_1 = r_{ab}\boldsymbol{x}_{ab} + r_1\boldsymbol{x}_{1_6} \quad \boldsymbol{v}_2 = r_{ab}\boldsymbol{x}_{ab} + r_2\boldsymbol{x}_{2_6}$$

We similarly define the vector $\boldsymbol{x}_{ba}$ using the sequences. In order for the 1-head, 1-layer model to solve all 2-hop induction instances, we must have vectors $\boldsymbol{x}_{ab}$, $\boldsymbol{x}_{ba}$, $\boldsymbol{x}_{1_6}$, and $\boldsymbol{x}_{2_6}$ that satisfy the following inequalities:

$$(r_{ab}\boldsymbol{x}_{ab} + r_1\boldsymbol{x}_{1_6}) \cdot (\boldsymbol{x}_a - \boldsymbol{x}_b) > 0, \quad (r_{ab}\boldsymbol{x}_{ab} + r_2\boldsymbol{x}_{2_6}) \cdot (\boldsymbol{x}_a - \boldsymbol{x}_b) < 0,$$
$$(r_{ba}\boldsymbol{x}_{ba} + r_1\boldsymbol{x}_{1_6}) \cdot (\boldsymbol{x}_a - \boldsymbol{x}_b) < 0, \quad (r_{ba}\boldsymbol{x}_{ba} + r_2\boldsymbol{x}_{2_6}) \cdot (\boldsymbol{x}_a - \boldsymbol{x}_b) > 0.$$

By Lemma 2, such vectors do not exist, implying the desired impossibility result. $\qquad\square$

## 4 ANALYSIS OF 2-HEAD TRANSFORMERS

**Theorem 3.** *For any directed graph with $n$ vertices, there exists a 2-head attention-only single-layer transformer model that can solve ESP with $O(n)$ dimensions and $O(1)$ precision.*

*Proof.* We provide a constructive proof by showing the existence of a set of model parameters that can solve ESP. Let $\mathcal{G} = (V, E)$ be a directed graph with vertex set $V$ of size $n$ and edge set $E$. Consider the input sequence $S_{1:4} = (u_1, v_2, i_3^*, \#_4)$ for an arc $(u, v) \in E$, where $i_3^*$ is the selector token. We set the embedding dimension to $d := n + 3$, with standard basis vectors $\{\boldsymbol{e}_1, \ldots, \boldsymbol{e}_d\} \subset \mathbb{R}^d$. The selector token $i_3^*$ is embedded using only a token embedding to $-M\boldsymbol{e}_{i^*}$, where $M$ is a constant that will be set suitably large later in the proof. The token $\#_4$ is embedded as $\boldsymbol{e}_3$. Each vertex $v \in V$ appearing at position $i \in \{1, 2\}$ is embedded as $\boldsymbol{x}_i = \boldsymbol{e}_{v+3} + \boldsymbol{e}_i$. The resulting embeddings form the input matrix $\boldsymbol{X} = [\boldsymbol{x}_1, \ldots, \boldsymbol{x}_4]^\top \in \mathbb{R}^{4 \times d}$.

Our model's final prediction for the correct endpoint vertex is given by

$$\hat{v} = \arg\max_{v \in V} \left[ \left( \sum_{j=1}^{2} \text{Softmax}\left(\#_4 \boldsymbol{A}_j \boldsymbol{X}^\top\right) \boldsymbol{X} \boldsymbol{V}_j \right) \boldsymbol{W}_0^\top \right]_v \qquad (\star)$$

where $[\cdot]_v$ denotes the logit corresponding to vertex $v \in \mathcal{G}$. We define $\boldsymbol{V}_1 = \boldsymbol{V}_2 = \boldsymbol{V}$, set $\boldsymbol{V}, \boldsymbol{W}_0$, and calculate $\boldsymbol{V}\boldsymbol{W}_0^\top$ as follows:

$$\boldsymbol{V} := \begin{bmatrix} \boldsymbol{O}_{3 \times n} & \boldsymbol{O}_{3 \times 3} \\ \boldsymbol{I}_n & \boldsymbol{O}_{n \times 3} \end{bmatrix} \qquad \boldsymbol{W}_0 := \begin{bmatrix} \boldsymbol{I}_n & \boldsymbol{O}_{n \times 3} \end{bmatrix} \qquad \boldsymbol{R} = \boldsymbol{V}\boldsymbol{W}_0^\top = \begin{bmatrix} \boldsymbol{O}_{3 \times n} \\ \boldsymbol{I}_n \end{bmatrix} \in \mathbb{R}^{(n+3) \times n}.$$

Given our construction of $\boldsymbol{V}$ and $\boldsymbol{W}_0$, the final logit vector inside $(\star)$ above simplifies to $(\alpha_1 + \alpha_2)\boldsymbol{X}\boldsymbol{R}$. Here $\boldsymbol{R}$ is a fixed selection matrix that zeros out the first two dimensions while keeping the remaining $n$ dimensions corresponding to the token embeddings for the vertices of our graph. Indeed, $\boldsymbol{X}\boldsymbol{R}$ is $[\boldsymbol{e}_u, \boldsymbol{e}_v, 0, 0]^\top$. Thus, the final prediction can be thought of as selecting which of $u$ or $v$ receives the largest weight under $\alpha_1 + \alpha_2$.

We define the attention matrices: in matrix $\boldsymbol{A}_j$, every element is 0 except for the element $(3, j)$, which is set to 1. Hence, we obtain

$$\#_4\boldsymbol{A}_1 = \begin{bmatrix} 1 & 0 & 0 & \cdots & 0 \end{bmatrix} \in \mathbb{R}^{1 \times d}, \qquad \#_4\boldsymbol{A}_2 = \begin{bmatrix} 0 & 1 & 0 & \cdots & 0 \end{bmatrix} \in \mathbb{R}^{1 \times d}.$$

Here $\#_4 A_1$ extracts the first row of $\boldsymbol{X}^\top$ (corresponding to position 1), while $\#_4 A_2$ extracts the second row of $\boldsymbol{X}^\top$ (corresponding to position 2).

**Evaluating** $\alpha_1 + \alpha_2$. Fix $i_3^* = 2$ (the case $i_3^* = 1$ is symmetric). For the two heads, we get

$$\#_4 \boldsymbol{A}_1 \boldsymbol{X}^\top = \boldsymbol{z}_1 = [1, 0, 0, 0], \qquad \#_4 \boldsymbol{A}_2 \boldsymbol{X}^\top = \boldsymbol{z}_2 = [0, 1, -M, 0].$$

$$\text{Softmax}(\boldsymbol{z}_1) = \tfrac{1}{e+3} \left[ e, 1, 1, 1 \right], \quad \text{Softmax}(\boldsymbol{z}_2) = \tfrac{1}{e+2+e^{-M}} \left[ 1, e, e^{-M}, 1 \right].$$

Thus, the first and second coordinates of $\alpha_1 + \alpha_2$ are $e/(e+3) + 1/(e+2+e^{-M})$ and $1/(e+3) + e/(e+2+e^{-M})$. By choosing $M$ sufficiently large, we ensure that the second coordinate is larger than the first coordinate by a constant that can be made arbitrarily close to $e - 1/((e+3)(e+2))$. Thus, the predicted token $\hat{v}$ is the letter $v$, ensuring that the correct endpoint vertex is selected. $\qquad \square$

**Corollary 2.** *There exists a $2$-head, $1$-layer, attention-only transformer model with **constant** embedding dimension $d = 5$ and precision $O(\log n)$ that solves ESP on any directed graph.*

*Proof.* We first give a proof for unbounded precision, and then extend the argument to $O(\log n)$ precision. Instead of a token embedding that embeds vertices as basis vectors in $\mathbb{R}^n$, we embed them as well-separated unit vectors in $\mathbb{R}^2$, and pad with a suitable positional embedding. Number the vertices $1$ through $n$, and define

$$\theta_\ell := \tfrac{2\pi\ell}{n}, \qquad \phi(v_\ell) := [\cos\theta_\ell \quad \sin\theta_\ell] \in \mathbb{R}^{1\times 2}.$$

For a vertex $v$ at position $i \in \{1, 2\}$, we create its embedding by adding the positional basis vector $\boldsymbol{e}_i \in \mathbb{R}^5$ to the padded vertex encoding, i.e.,

$$\boldsymbol{x}_i = \boldsymbol{e}_i + ([0, 0, 0] \oplus \phi(v)) \in \mathbb{R}^5.$$

Next, we update the value and output projection matrices $\boldsymbol{V}$ and $\boldsymbol{W}_0$ and calculate $\boldsymbol{R} = \boldsymbol{V}\boldsymbol{W}_0^\top$:

$$\boldsymbol{V} := \begin{bmatrix} \boldsymbol{O}_{3\times 2} & \boldsymbol{O}_{3\times 3} \\ \boldsymbol{I}_2 & \boldsymbol{O}_{2\times 3} \end{bmatrix}, \qquad \boldsymbol{W}_0 := \begin{bmatrix} \boldsymbol{E} & \boldsymbol{O}_{n\times 3} \end{bmatrix} \qquad \boldsymbol{R} = \boldsymbol{V}\boldsymbol{W}_0^\top \begin{bmatrix} \boldsymbol{O}_{3\times n} \\ \boldsymbol{E}^\top \end{bmatrix} \in \mathbb{R}^{5\times n},$$

where $\boldsymbol{E} \in \mathbb{R}^{n\times 2}$ is the token embedding matrix for all the vertices of the graph. We obtain that $\boldsymbol{X}\boldsymbol{R}$ is the $4 \times n$ matrix whose first row is the vector with $k$th coordinate being $\phi(k) \cdot \phi(u)$, second row is the vector with $k$th coordinate being $\phi(k) \cdot \phi(v)$ and remaining vectors being zero. (Here, for any vertex $k$, $\phi(k)$ is being viewed as a two-dimensional vector.) Note that $\phi(k) \cdot \phi(u) = \cos^2(\theta_u) + \sin^2(\theta_u) = 1$ for $k = u$ and is strictly less than $1$ for $k \neq u$. By the same argument as in the proof of Theorem 3, the coordinate of $\alpha_1 + \alpha_2$ that is maximized is the selector $i_3^*$. Consequently, the $n$-dimensional vector $(\alpha_1 + \alpha_2)\boldsymbol{X}\boldsymbol{R}$ has its maximum value in the coordinate corresponding to the vertex that appears in position $i_3^*$, yielding the desired result, assuming unbounded precision.

To obtain the result with $O(\log n)$ precision, we observe that when $k \neq u$, $\phi(k) \cdot \phi(u)$ equals $\cos(\theta_k)\cos(\theta_u) + \sin(\theta_k)\sin(\theta_u)$, which equals $\cos(\theta_k - \theta_u)$, which is at most $\cos(2\pi/n)$. By Taylor expansion, we have $\cos(2\pi/n) = 1 - \Omega(1/n^2)$. Therefore, it is sufficient to approximate $\cos(\theta_\ell)$ and $\sin(\theta_\ell)$ to an additive bound of $O(1/n^3)$. Thus, we can replace $\cos(\theta_\ell)$ and $\sin(\theta_\ell)$ by the nearest multiple of a rational number $1/r$, where $r$ is an integer, which is $O(n^3)$. This ensures that the coordinates with the largest value in the first two row vectors of $\boldsymbol{X}\boldsymbol{R}$ continue to be those corresponding to $u$ and $v$ respectively, leading to the desired prediction. Since all weights in the embeddings and the attention matrix are multiples of $O(\log n)$, the result follows. $\qquad \square$

## 5  EXPERIMENTS

We validate our theoretical results and probe the expressivity of attention heads for ESP.

**Experimental setup**. We use a decoder-only transformer with causal self-attention and no feedforward layers. Weights are tied between input embeddings and the output projection, and we retain residual connections and layer normalizations. We train with Adam and place no restrictions on embedding dimension or other hyper-parameters (training iterations, learning rate, batch size, etc.). All experiments were run on A100 and L4 GPUs via Google Colab.

**Datasets and metrics**. We generate transitive tournaments (the largest DAG for a given number of vertices) by labeling nodes $1$ through $|V|$ and including all arcs $(u, v)$ with $u < v$. To study cycles, we generate graphs with varying minimum feedback arc set (MFAS) sizes (see figures in

Appendix C.1). We define the accuracy as the fraction of correctly predicted arcs over all training sequences extracted from the graph.

**1-head models are sufficient for DAGs, insufficient for cycles; 2-head models solve all graphs.** Consistent with Theorem 1, a single-head model solves ESP on DAGs. On graphs with cycles, it approaches the global minimum only for simple cases with $|\text{MFAS}| = 1$, i.e., achieving 100% accuracy on the maximum acyclic subgraph (MAS) and 50% on the remaining (MFAS) arcs, essentially guessing heads/tails. Performance degrades for more complex graphs with $|\text{MFAS}| > 1$. (See Appendix C.1 for more details.) Adding a second head enables ESP to be solved on arbitrary directed graphs, consistent with Theorem 3 (see Figure 2(B)).

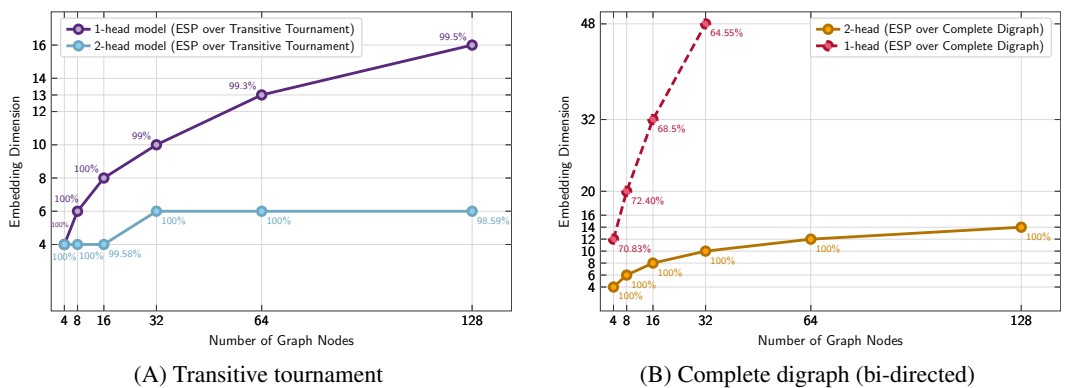

(A) Transitive tournament          (B) Complete digraph (bi-directed)

Figure 2: Accuracy plots for the best-performing models across different configurations.

**Analysis of embedding dimension.** We empirically study how the minimum dimension needed to solve ESP varies with graph size. In Figure 2(A), we observe that for DAGs, gradient-based optimization can find accurate constant-dimension 2-head models and nearly-accurate 1-head models with sublinear dimension (suggesting that the linear upper bound established in Theorem 1 can be improved). In Figure 2(B), we observe that for general graphs, gradient-based optimization finds accurate 2-head models with dimension that appears to grow logarithmically in the graph size, in contrast to the constant upper bound established in Corollary 2. Furthermore, 1-head models struggle on complete digraphs, and the gradient-based optimizer fails to find the best 1-head solution from Theorem 1. Since a complete digraph on $n$ vertices has $m = n(n-1)$ edges and a transitive orientation yields an acyclic subgraph with $m' = \binom{n}{2} = n(n-1)/2$, the bound gives error $= \frac{1}{2} - \frac{n(n-1)/2}{2\,n(n-1)} = \frac{1}{4}$, i.e., accuracy $\geq 3/4 = 75\%$. In practice, even with $d \gg n$, our 1-head models fail to get close to the 75% accuracy level on complete digraphs. Figure 2 reveals a clear separation: two narrow heads offer representational power that widening a single head cannot match. Additional experimental results, which analyze how bounded-width FFN components affect the accuracy of 1-head models for ESP, are provided in Appendix C.2.

## 6    LIMITATIONS, DISCUSSION AND CONCLUDING REMARKS

Our analysis focuses on a highly simplified setting: attention-only transformers with either one or two heads. While this abstraction allows us to isolate fundamental representational limits, it does not account for components such as feedforward layers, residual connections, or training dynamics. Despite simplifications, our results highlight structural bottlenecks in attention itself, independent of embedding size or numerical precision. The observed gap between a single-head and two-head models suggests the emergence of a natural hierarchy in transformer capabilities, providing a framework for analyzing model design choices beyond empirical scaling.

In sum, our theoretical and experimental findings suggest that attention heads act as discrete units of computational power, with each additional head expanding the class of solvable problems opening the door to a principled taxonomy of transformer architectures characterizing the boundaries of deeper or multi-head configurations. More broadly, our approach demonstrates how simple formal tasks like ESP can serve as testbeds for probing the fundamental limits of large-scale sequence models.

## ETHICS STATEMENT

This paper presents work whose goal is to advance the field of Machine Learning. There are many potential societal consequences of our work, none which we feel must be specifically highlighted here.

## REPRODUCIBILITY STATEMENT

This paper contains both theoretical results and experimental analyses. The main body of the paper and the appendix together contain all the assumptions and complete proofs of all claims. We have submitted the source code for all of our experimental analyses as supplementary material.

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

## A  ESP AS A SPECIAL CASE OF 2-HOP INDUCTION HEADS

**Induction Heads.**  Induction heads are a well-studied circuit-level phenomenon in transformer models (Elhage et al., 2021; Olsson et al., 2022), implemented by a pair of attention heads. They perform the following simple find-and-copy algorithm: given an input sequence $S = (s_1, \ldots, s_i, s_{i+1}, \ldots, s_T)$

1. find the last position $j < T$ where $s_j = s_T$,
2. predict the subsequent token $s_{j+1}$.

For example, given the input sequence $(a, \ldots, a, b, \ldots, a)$, the induction heads work together to predict $b$, since $b$ follows the previous occurrence of $a$. This framework generalizes out-of-distribution, since it executes an input-agnostic algorithm rather than memorizing fixed patterns.

**Two-Hop Induction.**  The two-hop induction problem, first noted in Sanford et al. (2024c), extends this mechanism by chaining together two of the above find-and-copy operations. For example, given the sequence

$$(a, \ldots, b, c, \ldots, a, b, \ldots, a),$$

the correct prediction is $c$, since the model must first retrieve the $b$ following $a_j$, and then output the token that followed the earlier occurrence of $b$, namely $c$. Thus, two-hop induction composes two one-hop induction steps, demonstrating how induction heads can be leveraged for more complex reasoning.

**Relation to ESP.**  We show that the **Endpoint Selection Problem (ESP)** is a special case of a 2-hop induction head problem and present a full proof of Corollary 1. Recall that in ESP, the input sequence is of the form

$$a\ b\ i\ \#,$$

where $a, b$ are endpoints of an arc, and $i \in \{1, 2\}$ is an indicator token. The target outputs are as follows:

$$a\ b\ 1\ \# \mapsto a, \quad a\ b\ 2\ \# \mapsto b, \quad b\ a\ 1\ \# \mapsto b, \quad \cdots$$

This input can be pre-processed into a form equivalent to a 2-hop induction instance. Specifically, we can pad each sequence as

$$1\ a\ 2\ b\ \#\ 1\ \#, \qquad 1\ a\ 2\ b\ \#\ 2\ \#, \qquad 1\ b\ 2\ a\ \#\ 1\ \#, \quad \cdots$$

In this representation, fixing the query token converts the task from a 1-hop to a 2-hop problem by "exhausting" a single hop. In particular, note that in the 2-hop induction head instances created, the tokens in positions 1, 3, and 5 are always the same. Hence, any transformer circuit makes the next-token prediction based on the tokens in positions 2 and 4 (each either $a$ or $b$), position 6 (either 1 or 2) and the preceding token (which is $\#$). This is identical to the transformer model that is solving an ESP instance. Therefore, one can design a transformer circuit for ESP by simply emulating one for 2-hop induction head. This implies that our impossibility result from Theorem 2 also extends to the 2-hop induction head problem.

We formalize the above argument, following the proof technique of Theorem 2. Consider the following two input sequences $(1_1, a_2, 2_3, b_4, \#_5, 1_6, \#_7)$ which must output $a$ and $(1_1, a_2, 2_3, b_4, \#_5, 2_6, \#_7)$ which must output $b$. As before, let the pre-softmax score for a token $s$ at position $p$ be denoted as $z_{s_p} = \#_7 \boldsymbol{A} x_{s_p}^\top$. For the two sequences, the attention weights are given by:

$$\text{Softmax}\big(\#_7 \boldsymbol{A} x_{1_1}^\top, \#_7 \boldsymbol{A} x_{a_2}^\top, \#_7 \boldsymbol{A} x_{2_3}^\top, \#_7 \boldsymbol{A} x_{b_4}^\top, \#_7 \boldsymbol{A} x_{\#_5}^\top, \#_7 \boldsymbol{A} x_{1_6}^\top, \#_7 \boldsymbol{A} x_{\#_7}^\top\big)$$
$$= \text{Softmax}(z_{1_1}, z_{a_2}, z_{2_3}, z_{b_4}, z_{\#_5}, z_{1_6}, z_{\#_7})$$

$$\text{Softmax}\big(\#_7 \boldsymbol{A} x_{1_1}^\top, \#_7 \boldsymbol{A} x_{a_2}^\top, \#_7 \boldsymbol{A} x_{2_3}^\top, \#_7 \boldsymbol{A} x_{b_4}^\top, \#_7 \boldsymbol{A} x_{\#_5}^\top, \#_7 \boldsymbol{A} x_{2_6}^\top, \#_7 \boldsymbol{A} x_{\#_7}^\top\big)$$
$$= \text{Softmax}(z_{1_1}, z_{a_2}, z_{2_3}, z_{b_4}, z_{\#_5}, z_{2_6}, z_{\#_7})$$

The final context vector for each sequence, $S_i$, is the weighted sum of the input embeddings, $\boldsymbol{v}_i = w_i \cdot \boldsymbol{X}_i$. The two context vectors can then be written as:

$$\boldsymbol{v}_1 = \left( \frac{e^{z_{1_1}} \boldsymbol{x}_{1_1} + e^{z_{a_2}} \boldsymbol{x}_{a_2} + e^{z_{2_3}} \boldsymbol{x}_{2_3} + e^{z_{b_4}} \boldsymbol{x}_{b_4} + e^{z_{\#_5}} \boldsymbol{x}_{\#_5} + e^{z_{\#_7}} \boldsymbol{x}_{\#_7}}{Z_1} \right) + \left( \frac{e^{z_{1_6}} \boldsymbol{x}_{1_6}}{Z_1} \right)$$

$$\boldsymbol{v}_2 = \left( \frac{e^{z_{1_1}} \boldsymbol{x}_{1_1} + e^{z_{a_2}} \boldsymbol{x}_{a_2} + e^{z_{2_3}} \boldsymbol{x}_{2_3} + e^{z_{b_4}} \boldsymbol{x}_{b_4} + e^{z_{\#_5}} \boldsymbol{x}_{\#_5} + e^{z_{\#_7}} \boldsymbol{x}_{\#_7}}{Z_2} \right) + \left( \frac{e^{z_{2_6}} \boldsymbol{x}_{2_6}}{Z_2} \right)$$

where $Z_i = \sum_{j \in S_i} e^{z_j}$ for $i \in \{1, 2\}$. We next define $\boldsymbol{x}_{ab}$, the attention-weighted sum over the non-indicator tokens:

$$\boldsymbol{x}_{ab} := \frac{e^{z_{1_1}} \boldsymbol{x}_{1_1} + e^{z_{a_2}} \boldsymbol{x}_{a_2} + e^{z_{2_3}} \boldsymbol{x}_{2_3} + e^{z_{b_4}} \boldsymbol{x}_{b_4} + e^{z_{\#_5}} \boldsymbol{x}_{\#_5} + e^{z_{\#_7}} \boldsymbol{x}_{\#_7}}{e^{z_{1_1}} + e^{z_{a_2}} + e^{z_{2_3}} + e^{z_{b_4}} + e^{z_{\#_5}} + e^{z_{\#_7}}}$$

Using these definitions, we can express each of the final context vectors as a convex combination of the newly defined vectors and the indicator tokens.

$$\boldsymbol{v}_1 = \left( \frac{e^{z_{1_1}} + e^{z_{a_2}} + e^{z_{2_3}} + e^{z_{b_4}} + e^{z_{\#_5}} + e^{z_{\#_7}}}{Z_1} \right) \boldsymbol{x}_{ab} + \left( \frac{e^{z_{1_6}}}{Z_1} \right) \boldsymbol{x}_{1_6} := r_{ab} \boldsymbol{x}_{ab} + r_1 \boldsymbol{x}_{1_6}$$

$$\boldsymbol{v}_2 = \left( \frac{e^{z_{1_1}} + e^{z_{a_2}} + e^{z_{2_3}} + e^{z_{b_4}} + e^{z_{\#_5}} + e^{z_{\#_7}}}{Z_2} \right) \boldsymbol{x}_{ab} + \left( \frac{e^{z_{2_6}}}{Z_2} \right) \boldsymbol{x}_{2_6} := r_{ab} \boldsymbol{x}_{ab} + r_2 \boldsymbol{x}_{2_6}$$

We similarly define the vector $\boldsymbol{x}_{ba}$ using the sequences $(1_1, b_2, 2_3, a_4, \#_5, 1_6, \#_7)$, which must output $b$, and $(1_1, b_2, 2_3, a_4, \#_5, 2_6, \#_7)$, which must output $a$. As we argue at the end of the proof of Theorem 2, Lemma 2 implies that there do not exist vectors $\boldsymbol{x}_{ab}$, $\boldsymbol{xba}$, $\boldsymbol{x}_{1_6}$, and $\boldsymbol{x}_{2_6}$ satisfying the four inequalities:

$$(r_{ab} \boldsymbol{x}_{ab} + r_1 \boldsymbol{x}_{1_6}) \cdot (\boldsymbol{x}_a - \boldsymbol{x}_b) > 0,$$
$$(r_{ab} \boldsymbol{x}_{ab} + r_2 \boldsymbol{x}_{2_6}) \cdot (\boldsymbol{x}_a - \boldsymbol{x}_b) < 0,$$
$$(r_{ba} \boldsymbol{x}_{ba} + r_1 \boldsymbol{x}_{1_6}) \cdot (\boldsymbol{x}_a - \boldsymbol{x}_b) < 0,$$
$$(r_{ba} \boldsymbol{x}_{ba} + r_2 \boldsymbol{x}_{2_6}) \cdot (\boldsymbol{x}_a - \boldsymbol{x}_b) > 0.$$

This establishes that there is no 1-head, 1-layer attention-only transformer model for 2-hop induction head, even with unbounded dimension and unbounded precision.

## B  NP-COMPLETENESS

We demonstrate the intractability of even approximating the minimum error of a 1-head model for ESP on general directed graphs.

But first we state a corollary that follows directly from the theorems in Section 3. We remind the reader that MAS stands for Maximum Acyclic Subgraph, MFAS for Minimum Feedback Arc Set and for any directed graph with $m$ arcs, $|\text{MAS}| + |\text{MFAS}| = m$.

**Corollary 3.** *For any integer $n$ and any directed graph $G$, which has $n$ vertices, $m$ edges, and an acyclic subgraph with $|\text{MAS}|$ edges, there exists a 1-head transformer model with embedding dimension $n + 1$ that incurs an error exactly equal to $1/2 - |\text{MAS}|/(2m)$ for ESP on $G$.*

**Theorem 4.** *Finding a 1-head minimum error model for ESP is NP-complete.*

*Proof.* For the sake of formalization, let us define 1H-ESP to be the decision problem: given directed graph $G$ and error $\epsilon$ accept if and only if there is a 1-head model which achieves error at most $\epsilon$ on $G$. As always, $n$ and $m$ stand for the number of vertices and arcs, respectively, of $G$. The reduction is from known NP-hard problem MAS (Maximum Acyclic Subgraph) Karp (1972); Garey & Johnson (1979), with the following definition as a decision problem: given a directed graph $G$ and number $m'$, accept if and only if there is a DAG with at least $m'$ arcs that is a subgraph of $G$. We prove NP-completeness of 1H-ESP by first showing that it is in NP and then showing that it is NP-hard by a reduction from MAS.

To see that 1H-ESP is in NP, guess any DAG contained in G with at least $m'$ arcs and then follow the construction in the proof of Theorem 1 to obtain a certificate (NP-witness) that the error is $1/2 - m'/(2m)$.

Next, to see NP-hardness of 1H-ESP here is the straightforward reduction. Given an instance $\langle G, m' \rangle$ of MAS we transform it into instance $\langle G, 1/2 - m'/(2m) \rangle$. By Corollary 3 we know that $m'$ is the size of the largest DAG if and only if the minimum error achievable is $1/2 - m'/(2m)$, and hence it follows that the MAS instance is accepted if and only if the 1H-ESP instance to which it is reduced is accepted. $\qquad\square$

We now strengthen the NP-completeness to an APX-hardness.

**Theorem 5.** *The minimum error of the best 1-head model for ESP cannot be approximated to a factor better than 1.3606.*

*Proof.* The proof follows in straightforward fashion from the APX-hardness of MFAS, the Minimum Feedback Arc Set problem (Kann, 1992), which is known not to be approximable to a factor better than 1.3606, unless P = NP. Observe that the error in Corollary 3, $1/2 - |\text{MAS}|/(2m)$, is the same as $|\text{MFAS}|/(2m)$ and thus the inapproximability factor carries over exactly. $\qquad\square$

**Corollary 4.** *Gradient descent cannot compute (even approximate) the global minimum (to better than a factor of 1.3606) of ESP for 1-head models in polynomial time, unless P = NP.*

## C  ADDITIONAL DETAILS FOR EXPERIMENTS WITH DIRECTED GRAPHS

### C.1  GRAPH VISUALIZATIONS FOR ESP EXPERIMENTS

Figure 3 presents the family of directed graphs (with cycles) over which we analyze the 1-head and 2-head transformer models. Empirically, the best accuracies were obtained after more than 50 runs with hyperparameter tuning for Figure 3(B) and Figure 3(C), and after more than 30 runs for Figure 3(A) and Figure 3(D), within 10k–20k training iterations each.

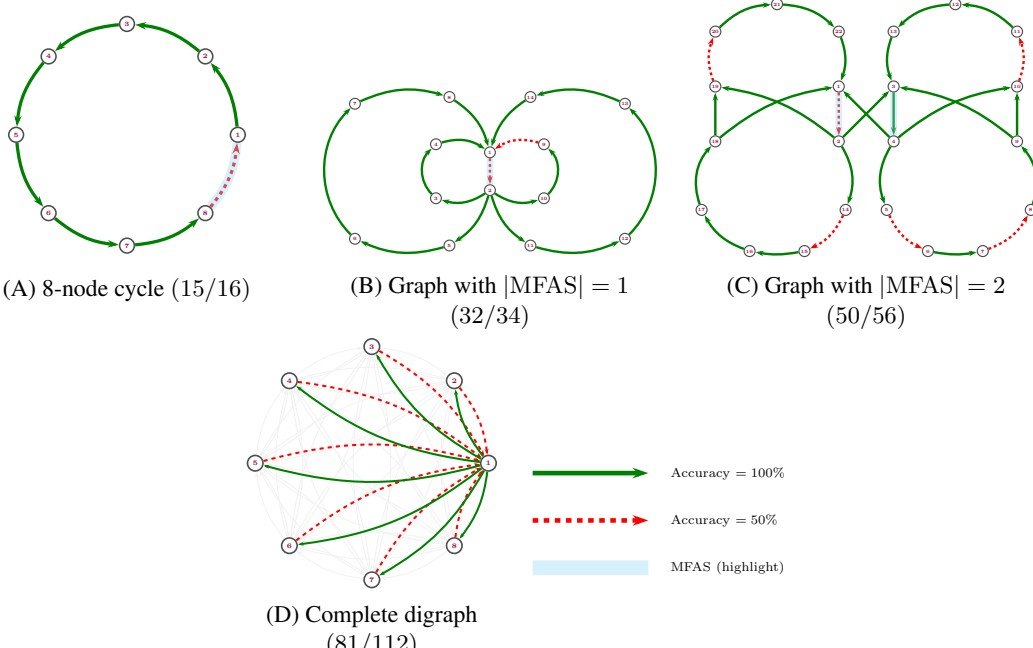

(A) 8-node cycle (15/16)

(B) Graph with $|\text{MFAS}| = 1$ (32/34)

(C) Graph with $|\text{MFAS}| = 2$ (50/56)

(D) Complete digraph (81/112)

Figure 3: Prediction accuracies for the best-performing 1-head model for different graphs. Note that (D) only shows edges adjacent to vertex 1, and MFAS highlight only applies to (A), (B), and (C).

## C.2 BOUNDED-WIDTH FFN EXPERIMENTS FOR ESP

To further investigate how bounded-width feed-forward (FFN) components affect the representational limits of shallow transformer models, we conducted an additional experiment measuring the smallest model size required to solve ESP over complete directed graphs of varying sizes. For each graph size, we identify the minimum-parameter configuration of (i) a 2-head attention-only transformer and (ii) a 1-head transformer augmented with a bounded-width FFN that achieves near-perfect accuracy.

In this experiment, the parameter count refers to the total number of trainable parameters used by each respective model, including all attention matrices and (when present) the parameters in the FFN block.

Figure 4 summarizes the results. The key takeaway is that a 1-head transformer equipped with a bounded-width FFN can match the expressivity of a 2-head attention-only transformer when using a comparable number of total parameters.

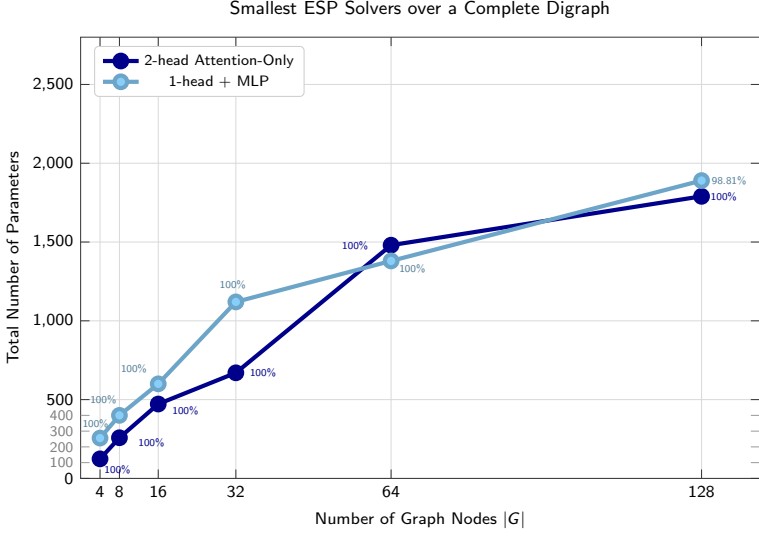

Figure 4: Number of parameters of the smallest models that solve ESP over complete digraphs.

## D    LLM USAGE

We used LLMs to help identify related work, transcribe handwritten equations into LaTeX, and polish writing and grammar.

