# OpenReview forum: "Two (narrow) heads are better than (an arbitrarily wide) one"
_ICLR.cc/2026/Conference — ICLR 2026 Poster_

### Official Review · Reviewer_kCWe · 2025-10-25

**Soundness:** 4
**Presentation:** 3
**Contribution:** 3
**Rating:** 4
**Confidence:** 4

**Summary:**

The authors analyze a simplified transformer model (1-layer, 1- or 2-head, attention-only) on the endpoint selection problem (ESP). Given a graph, ESP is a distribution of problem instances over the edges in the graph. Given a uniformly sampled edge $(u, v)$, the task is to select either $u$ or $v$, based on whether the next token is $1$ (select $u$) or $2$ (select $v$). The authors show that a 1-head, 1-layer, attention-only transformer can solve ESP in $n$-node DAGs with embedding dimension $n$ (Theorem 1 is slightly more general, applying to directed graphs with large enough acyclic subgraphs, with accuracy degrading as the subgraph shrinks). In contrast, the authors show that 1-head, 1-layer attention-only transformers cannot (exactly) solve ESP in cyclic graphs, and finding the best-approximating transformer is NP-complete. Additionally, they show 2-head models can solve ESP for any directed graph. Experimental results support the theory, but indicate bound are loose.

**Strengths:**

1. Well-written paper overall and fairly clear, modulo some minor issues noted below
2. Nice combination of theoretical results and experiments.
3. Topic (analysis of transformers) is important and relevant. Significance is mixed; narrow problem and model, but the area is still under-explored, which I think overrules the narrowness
4. Overall quality is high, what I checked of the proofs seems sound

**Weaknesses:**

1. The connection to induction heads felt shoe-horned in; the connection to index is much clearer, which highlights a critical piece of missing related work [1], whose results need to be discussed, as they at first seem to contradict the results here (see question 2)
2. Given the results of [1] on Index for 1-layer transformers with FFN, the importance of this analysis of 1-layer attention-only transformers needs to be justified


### Overall evalution
I actually think this is a nice paper, but the relationship to [1] really needs to be addressed before publication, hence my initial low score. I am open to recommending acceptance if the authors can address these concerns in rebuttal/discussion and amend the paper accordingly.

**Questions:**

### Suggestions/comments
- In 1.1, the definition of ESP is confusing, as the first sentence of 1.1 says it's a problem in a graph and the second sentence has no graph in the input to the problem. Only after carefully thinking about the third sentence (where the graph affects the distribution of instances generated) does the problem begin to make sense. Clarifying sooner that ESP in a graph is not one input-out pair but a distribution over input-output pairs would help.
- [1] should be cited and its Index result discussed wrt ESP (see question 2).
- in the transformer model definition, all terms should be named in the text ($E$, $P$, $X$, $Q$, $K$, $V$), even if they are standard.
- the style of plot in figure 2 (with accuracy shown as labels only and a hyperparameter on the y axis) is a bit odd, as it doesn't show the accuracy of models with other embedding dimensions. (e.g., how do we know there isn't a smaller one-head model with accuracy 100%, from that plot?) I'm not sure how to fix this best... maybe, for size $n$, also show the accuracy of the next-smaller model, so we see that shrinking it worsens performance?
- Theorem 5 is only mentioned in the introduction; discussing it in the results section as well would be better. And theorem 4 is never mentioned! This is a nice result to complement theorem 2, which otherwise might feel overly restrictive with its exact solution requirement.
 - Great title. But I would suggest adding something about edge endpoint selection.
 - The related work does a reasonable job overall, but there are some other papers that could have been mentioned [3-9]. See [2] for a nice survey. Of course, no one paper needs a totally exhaustive related work (unless it's a survey), but these might be useful places for the authors to look--and at least pointing to the survey would be useful for readers. (And I was surprised to see no cited papers by several important people in this area, like Strobl, Hahn, Chiang, and Bhattamishra.) For a different perspective on several relevant problems like Index and graph reachability, see also [10].


### Questions
1. The connection between 2-hop induction heads feels quite tenuous, and it's not clear to me that we can say ESP reduces to 2-hop induction heads. This "reduction" requires transforming the input, so it's not clear that it's valid to say that "a circuit capable of two-hop induction can directly solve ESP," (p. 13) as that circuit is not capable of performing the input transformation. If we allow arbitrary input transformations, then we could "reduce" any problem to another by placing the answer at the end of the new input. In Turing machine polynomial-time reductions, this style of issue is addressed by ensuring the TM itself can perform the input transformation in polynomial time. But here, it's not clear how to make the transformer do the input transformation to then apply the 2-hop induction heads circuit. How can we make this notion of reduction well-defined?
2. Instead of drawing a parallel between an ESP instance and 2-hop induction heads, the much more natural analogy is to the Index problem: given an input and an index, retrieve the item at the index. This is exactly what ESP is. But [1], Theorem 1 shows that there is a 1-layer transformer with log width and precision that solves Index, and as far as I can tell their construction has only a single head, which seems to contradict the authors' Theorem 2. Their log-width also seems better than the O(n) embedding dimension of Theorem 1. What explains the differences in the results? Ah, I think it's that the results here are for attention-only transformers, whereas [1] uses a (small) feed-forward network. Does their construction become impossible with just a linear layer + softmax instead of the FFN? My intuition is that index still feels possible by attending to the position embedding.... (then again, the logic of theorem 2 makes sense, but it's worth identifying exactly what about their construction fails to translate)
3. Defining ESP as including the query token seems unnecessary; why not just say the input is $(u, v, i)$?
4. Given that the results of [1] on Index are for a more realistic transformer model than attention-only, why are the results here for attention-only transformers on a restricted Index problem still valuable? (I don't claim they are not, but this needs to be justified)


### References
[1] Bhattamishra, S., Hahn, M., Blunsom, P., & Kanade, V. (2024). Separations in the representational capabilities of transformers and recurrent architectures. Advances in Neural Information Processing Systems, 37, 36002-36045.

[2] Strobl, L., Merrill, W., Weiss, G., Chiang, D., & Angluin, D. (2024). What formal languages can transformers express? a survey. Transactions of the Association for Computational Linguistics, 12, 543-561.

[3] Hahn, M. (2020). Theoretical limitations of self-attention in neural sequence models. Transactions of the Association for Computational Linguistics, 8, 156-171.

[4] Hahn, M., & Rofin, M. (2024, August). Why are Sensitive Functions Hard for Transformers?. In Proceedings of the 62nd Annual Meeting of the Association for Computational Linguistics (Volume 1: Long Papers) (pp. 14973-15008).

[5] Chiang, D. (2024). Transformers in uniform TC $^ 0$. arXiv preprint arXiv:2409.13629.

[6] Strobl, L. (2023). Average-hard attention transformers are constant-depth uniform threshold circuits. arXiv preprint arXiv:2308.03212.

[7] Bhattamishra, S., Ahuja, K., & Goyal, N. (2020, November). On the Ability and Limitations of Transformers to Recognize Formal Languages. In Proceedings of the 2020 Conference on Empirical Methods in Natural Language Processing (EMNLP) (pp. 7096-7116).

[8] Chiang, D., & Cholak, P. (2022, May). Overcoming a Theoretical Limitation of Self-Attention. In Proceedings of the 60th Annual Meeting of the Association for Computational Linguistics (Volume 1: Long Papers) (pp. 7654-7664).

[9] Strobl, L., Angluin, D., Chiang, D., Rawski, J., & Sabharwal, A. (2025). Transformers as transducers. Transactions of the Association for Computational Linguistics, 13, 200-219.

[10] Schnabel, T., Tomlinson, K., Swaminathan, A., & Neville, J. (2025). Lost in transmission: When and why LLMs fail to reason globally. arXiv preprint arXiv:2505.08140.

---

> ### Author Response · Authors · 2025-11-19
> **Response to kCWe**
>
> Thank you for your very thorough review of our paper and the many suggestions you have made to improve it. In the following, we respond to your suggestions and four specific questions. We also include a more detailed comparison with [1] in a separate comment.
>
> **Question 1 (Connection to induction head):** You are correct that we had not provided a **reduction** from ESP to 2-hop induction heads under the standard notion of reduction in complexity theory. The point we were aiming to make in our submission by presenting the transformation of an ESP instance to a 2-hop induction head instance is that **our 1-head impossibility proof for ESP also extends to the 2-hop induction head problem**.  In the revision, we have added Corollary 1 (to Theorem 2) with a proof sketch establishing that there is no 1-head, 1-layer attention-only transformer model that solves $2$-hop induction heads, and have also included a full proof in the appendix. (Read at your own peril :-) - in technically precise terms, we show a reduction in vector-world, i.e, from a superset of ESP solutions to a superset of 2-hop induction head solutions.  So, this is not technically a reduction in the transformer-world but since our impossibility of existence of a solution to ESP applies to the vector-world, which is a superset of ESP transformer solutions, the vector-world reduction implies the non-existence of solutions to 2-hop induction head in the transformer world).
>
> **Question 2 (Connection to the Index problem in [1]):** Thank you for drawing the connection to the Index problem.  The comparison of ESP to Index is apt and important to discuss, and we have included a discussion on the two problems in the revision.  There are two critical differences.  First, the query token makes the two problems different.  Indeed, our impossibility argument for 1-head 1-layer attention-only transformers (with unbounded width and precision) also extends to the variant of Index, which includes a query token.  This is a sharp separation from the log-width, log-precision 1-head 1-layer argument in [1], and also indicates the fragility of several upper bound results.  As you mention, the model of [1] also includes an FFN layer, though it is not strictly necessary (using the ideas you allude to in your review, also see below). A second significant difference between the two formulations is that our ESP problem framework is defined over graphs and captures the limitations of one-head transformers and capability of two-head transformers to resolve the problem defined over different graph classes.  In particular, the inapproximability result indicates that finding a 1-head model that approximates accuracy to within a 1.3606 factor is intractable.  This enables us to identify limitations in the representation of the model and the intractability of finding weights that lead to a desired level of accuracy.
>
> **Question 3 (Use of the query token):** We have defined ESP with the query token for multiple reasons (as also elaborated in the Metaresponse). First, it is perhaps one of the simplest problems that leads to the impossibility and intractability results with attention-only models that we have derived.  Second, it allows us to extend the lower bounds to a range of other problems, including the two-hop induction head problem (which does not have a dedicated query token) as well as variants of the Index problem (as discussed above).  Third, it enables a natural generalization to hierarchies of problems that will test the capabilities of multi-layer multi-head transformers; for instance, the ESP problem can be easily composed with one another (e.g., using direct products) to obtain hypergraph variants, multi-way variants (with multiple indexes) and multi-hop versions (similar to $k$-hop induction heads).
>
> **Question 4 (Value of our results in the context of the Index problem):** We recognize the importance of the contributions of [1].  While their results on Index use more realistic models, our contributions remain valuable because our results are **unconditional**—holding for unbounded dimension and precision, unlike prior conditional bounds, and we establish a direct connection between transformer expressivity and computational complexity.  Furthermore, our formalization of ESP, though superficially similar to Index, exhibits fundamentally different behavior tied to graph-theoretic properties.  The connection to 2-hop induction heads (Appendix A) provides context for how ESP fits within the broader transformer circuit literature.
>
> We appreciate the extensive list of suggestions for further literature to make reference to in our paper. We have selected those papers that seem to us to be the most directly relevant to our work and have included a discussion in related work. We hope to have clarified the notation and addressed your suggestions satisfactorily in our revision. In the light of these changes, we hope you consider your support for our paper.

---

> ### Author Response · Authors · 2025-11-19
> **More comments on the relationship to [1]**
>
> We agree that the Index lookup result of Bhattamishra et al. [1] is important, and we clarify below why our attention-only analysis remains valuable and distinct.
>
> **First**, as you correctly observed, a key difference is that in the Index problem the *index token itself serves as the querying token*, whereas in ESP the selector token cannot induce a **selector-dependent key–query alignment** that switches between the two endpoints of an edge. This input-dependent shift in attention is exactly what a single attention head cannot realize for ESP over cyclic graphs.  Our impossibility and intractability results formalize these limitations.
>
> **Second**, as you allude to in the review, because the querying token contains the index, one can in fact implement a solution to the Index problem even in an *attention-only* 1-layer transformer with **constant width** (e.g., `d = 4`) and **logarithmic precision**, using the same “unit circle’’ idea as in Corollary 2 of our paper. Tokens and positions can be embedded using
> $$\phi(v_\ell) = [\cos \theta_\ell, \sin \theta_\ell], \text{ where } \theta_\ell = \frac{2\pi\ell}{n}$$
> and concatenated to obtain a 4-dimensional representation. Similar to their analysis, setting `W_Q` to extract the positional unit vector of the *index token*, and `W_K` to extract the positional component of each `s_i` (while zeroing out the index token), causes the softmax to shift the weights to the correct position. Instead of the FFN used in [1] to decode binary encodings, one may use the same linear “recovery’’ matrix `R = V W_0` from our Theorem 3 to project the attended value back to the token embedding, after which `argmax` recovers the desired sequence element.
>
> **Thus, there is no contradiction:** Index becomes solvable with constant width even without an FFN because the querying token encodes the entire index, whereas ESP requires instance-dependent attention behavior that a single attention head cannot implement. Our result isolates a limitation of **attention itself**, independent of width or precision, and therefore complements the findings of [1].

---

> ### Comment · Reviewer_kCWe · 2025-11-19
>
> Thanks for the detailed reply! The importance of the query token to these results is interesting, and that itself is revealing about the weaknesses of a single attention layer. In terms of framing, it does feel a bit awkward to define the problem in a slightly odd way that makes it harder (with the query token) and then discuss the hardness of the problem.
>
> I think the logic of the paper might benefit from highlighting the query token more as the source of difficulty (vs the tractability of index), and then arguing why analyzing the problem with the query token is interesting. E.g., it helps us understand two-hop induction heads, and in real inputs to transformers, we often have a "query token" (the \<eos\> token denoting the end of a prompt or even just a final instruction like "the answer is: " --- then again, in practice we have more than one layer).
>
> Given the updates, I'm happy to increase my score.

---

> > ### Author Response · Authors · 2025-11-20
> >
> > We appreciate your insight about reframing and increased support of the paper! We will incorporate your suggestions in our subsequent revisions, where we also hope to address any additional comments from other reviewers.

---

### Official Review · Reviewer_LMjh · 2025-10-28

**Soundness:** 3
**Presentation:** 3
**Contribution:** 2
**Rating:** 4
**Confidence:** 3

**Summary:**

In this article, the authors present new results about the expressivity of transformers.

The main contributions are:

- the introduction of a problem as testbed: the Endpoint Selection Problem (ESP) to study the expressive power of Transformers

- using this problem, the authors show a separation between one-head and two-heads transformers,  in the sense that (i) some instances of this problem cannot be solved exactly, for any one-head transformer, (ii) for any instance of this problem, there is a two-head transformer that solves it with zero error.

- an NP-complete result for the approximation of the best single-head model to minimize error on the arcs of graph.

- illustration of their results via experiments

**Strengths:**

- At a high-level, the results are appealing and interesting for the science of transformers, which is a fundamental part of all the recent progress in AI.

- Technically speaking the results hold for any dimension of the one-head, and there are quantitative bounds for the two-head, which gives a clean separation.

- The authors give a good overview of the existing results in the literature of the expressivity of transformers.

**Weaknesses:**

- The authors do not elaborate and give convincing arguments on why ESP is interesting to study in the first place (except some mention of reductions in the beginning of page 4, but then it is unclear why focus not on the other reductions). Since the authors are introducing a first in-depth study via the ESP, this is an important part missing to the article in my opinion.

- Related to the previous point, despite the clean seperation result, the nature of the result and significance feels limited. (to the authors: please feel free to clarify during rebuttal).

**Questions:**

- In the Appendix A., the example given for Induction Heads, do the authors mean: (a,a,a, .., a, b, a, ..., a), where b is in i-th position? Similar question for the Two-Hop induction.

- Could the authors elaborate on their choice of the ESP for the separation, in particular what lead them to consider this problem?

---

> ### Author Response · Authors · 2025-11-19
> **Response to LMjh**
>
> Thank you very much for the constructive feedback on our paper and for the positive assessment of our contributions, particularly the clean separation between one-head and two-head transformers and the generality of our lower bound. We would like to address the weaknesses you pointed out along with your questions below. In addition, we would like to draw your attention to the revised version of the paper with the edits made during the discussion phase in red.
>
> **Weakness 1/Question 2 (The relevance of ESP):** We have chosen ESP for its simplicity and the separation result that can be derived using it. Please see Theme 1 in our metaresponse for a detailed explanation of our choice of problem. Furthermore, in our revision, we have provided more motivation for ESP in Sections 1.1 and 1.2 of the paper.
>
> **Weakness 2 (The significance of our results):** In addition to the points raised in Theme 1 of our top-level comment, we would like to kindly ask you to refer to Theme 3 in the comment. We have revised our related work section to better situate it in the literature of the theoretical study of transformers, showing that both our model type (attention-only transformers) and problems of a similar type (such as the Index problem of Bhattamishra et al. [1] pointed out by Reviewer kCWe) have been studied, with our work opening new avenues of inquiry and adding novel theoretical results that, among other contributions, establish a connection between transformer expressivity and computational complexity.
>
> **Question 1 (Notation in Appendix A):** Thank you very much for pointing out the notation here. We have updated it in our revised version and hope to have clarified this issue.
>
> We hope that we have answered your questions satisfactorily and that you will consider raising your support for our paper, or help us understand how we might better address your concerns.
>
> [1] Bhattamishra, S., Hahn, M., Blunsom, P., & Kanade, V. (2024). Separations in the representational capabilities of transformers and recurrent architectures. Advances in Neural Information Processing Systems, 37, 36002-36045.

---

### Official Review · Reviewer_UJCu · 2025-10-31

**Soundness:** 3
**Presentation:** 3
**Contribution:** 3
**Rating:** 6
**Confidence:** 3

**Summary:**

This paper investigates the representational limits of attention-only transformers through the Endpoint Selection Problem (ESP), where models must select endpoints of directed graph arcs based on an indicator token. The authors establish dimension- and precision-independent impossibility results: no 1-head, 1-layer attention-only transformer can solve ESP on graphs with cycles, while 2-head models can solve ESP on arbitrary graphs with $\mathcal{O}(n)$ dimension and $\mathcal{O}(1)$ precision, or $\mathcal{O}(1)$ dimension and $\mathcal{O}(\log n)$ precision. For directed acyclic graphs (DAGs), 1-head models suffice with $\mathcal{O}(n)$ dimension. The paper proves it is $\mathsf{NP}$-complete to approximate the minimum error achievable by 1-head models and validates findings experimentally, demonstrating that gradient descent reliably finds 1-head solutions for DAGs and 2-head solutions for cyclic graphs.

**Strengths:**

1. **Strong theoretical contributions with unconditional lower bounds**. The paper's main result (Theorem 2) establishes impossibility for 1-head models on graphs with cycles without any restrictions on embedding dimension or precision, which is a significant advance over prior conditional lower bounds by Peng et al. (2024) and Sanford et al. (2024c) that required bounded dimension and precision. The geometric proof technique using Lemmas 1 and 2 is elegant and provides clear intuition through the half-space analysis shown in Figure 1. The matching upper bounds (Theorems 1 and 3) with constructive proofs and the computational hardness result (Theorem 5) create a complete characterization of the problem.

2. **Clean problem formulation with practical relevance**. ESP provides a well-motivated testbed that connects to the important induction head phenomenon studied in transformer interpretability (Appendix A demonstrates ESP as a special case of 2-hop induction heads). The problem is simple enough to admit rigorous analysis yet expressive enough to capture fundamental graph traversal and selection primitives underlying many reasoning tasks. The experimental validation on transitive tournaments and complete digraphs (Figure 2) demonstrates practical implications of the theoretical results.

3. **Comprehensive complexity-theoretic analysis**. Beyond expressivity results, the paper proves that finding optimal 1-head solutions is computationally intractable. Theorem 4 shows NP-completeness of finding minimum-error models, while Theorem 5 strengthens this to APX-hardness with an inapproximability factor of 1.3606 (lines 695-721). This is achieved through reduction from the Maximum Acyclic Subgraph problem, and Corollary 3 directly implies "gradient descent cannot compute (even approximate) the global minimum...in polynomial time, unless P = NP." These results provide crucial theoretical justification for why the optimization gaps observed in experiments (Section 5) may be fundamental rather than artifacts of training procedures.

**Weaknesses:**

1. **Limited scope due to attention-only restriction**. The paper focuses exclusively on attention-only transformers without feed-forward networks (FFN), as acknowledged in lines 065-067: "Since FFN with a single layer is already a universal approximator (Hornik et al., 1989), in this paper we focus on attention-only transformers." While this isolation enables clean theoretical analysis, it significantly limits practical relevance since real transformers crucially rely on FFN layers.

2. **Incomplete coverage of related work on transformer depth and circuit complexity**. The related work section (lines 136-150) discusses some depth-related results, mentioning that "transformers with constant depth, context length n, and precision O(log n) can be simulated by uniform constant-depth threshold circuits" (Merrill & Sabharwal, 2023) and that "log-depth transformers can solve graph connectivity" (Merrill & Sabharwal, 2024a), but omits several highly relevant recent works establishing fundamental complexity bounds for transformer architectures. Chen et al. (2024) [1] analyze circuit complexity bounds for RoPE-based transformer architectures, Cao et al. (2025) [2] show the circuit complexity bounds for vision transformers and their application, and Chiang (2025) [3] proves transformers operate in uniform $\mathsf{TC}^0$. These works are particularly relevant because RoPE is the dominant positional encoding in modern LLMs while this paper uses basic additive positional embeddings, and the circuit complexity perspective provides complementary lower bounds that could strengthen the impossibility results.

### References

[1] Bo Chen, Xiaoyu Li, Yingyu Liang, Jiangxuan Long, Zhenmei Shi, Zhao Song, Jiahao Zhang. “Circuit Complexity Bounds for RoPE-based Transformer Architecture”. EMNLP 2025.

[2] Yang Cao, Yubin Chen, Zhao Song, Jiahao Zhang. “Towards High-Order Mean Flow Generative Models: Feasibility, Expressivity, and Provably Efficient Criteria”. arXiv:2508.07102.

[3] David Chiang. “Transformers in Uniform TC0”. TMLR 2025.

**Questions:**

1. Does the 1-head impossibility result (Theorem 2) extend to transformers with feed-forward layers?
2. Can your geometric proof framework establish similar hierarchies for other architectural dimensions beyond head count?

---

> ### Author Response · Authors · 2025-11-19
> **Response to UJCu**
>
> We thank you for your positive review of our paper and your pointers to other papers in this area that are relevant to our work. We would like to address the weaknesses indicated in the review:
>
> **Regarding Weakness 1 (Limited scope due to attention-only restriction):** we are focussing on attention-only models in our paper due to their tractability and to isolate the role of attention, both of which could be helpful in establishing hierarchies. Please see the second theme in our metaresponse for a longer discussion on these points. In addition, the focus on attention-only transformers has been a consistent feature of research in this area (as evidenced by multiple papers cited in our literature review). Despite the narrower scope of our paper than work on transformers with FFNs, we believe the attention mechanism itself is a worthwhile area of rigorous study and hope that you agree that we have made a contribution in that respect.
>
> **Regarding Weakness 2 (Incomplete coverage of related work on transformer depth and circuit complexity):**  We thank you for suggestion previous works that we have missed in our literature review. We have addressed this issue in the revision uploaded concurrently with this rebuttal. Please let us know if there are further edits in this area you would like to see.
>
> **Q1.** Does the 1-head impossibility result (Theorem 2) extend to transformers with feed-forward layers?
>
> > **A1.** No, Theorem 2 does not extend to transformers with feed-forward layers. We are currently conducting experiments and are exploring this research direction further, but our current proof does not apply to transformers with feed-forward layers.
>
> **Q2.** Can your geometric proof framework establish similar hierarchies for other architectural dimensions beyond head count?
>
> > **A2.** We present the geometric approach to proving our result in the hope that it will lead to the establishment of similar results for dimensions such as width or depth (possibly with an application to transformers with feed-forward networks). However, we currently do not concretely which hierarchies will be established using this approach and we hope that our paper contributes to further inquiry along these lines.
>
>
> Please let us know if there are further changes or clarifications you would like to see to this paper. If there are no further points, we hope you consider raising your score.

---

### Official Review · Reviewer_WJsn · 2025-11-01

**Soundness:** 3
**Presentation:** 3
**Contribution:** 2
**Rating:** 4
**Confidence:** 3

**Summary:**

The authors provide a theoretical analysis of single-layer attention-only transformers and their ability to solve a simple problem called the Endpoint Selection Problem (ESP). This problem is interesting and motivated by the fact that it is foundational in many reasoning tasks. They show that single-layer transformers with a single attention head are not able to solve this problem in its general form even with unbounded model dimension. On the other hand, the authors show that the same model with two attention heads can easily solve the problem with limited model dimension and precision. They provide experimental results that further corroborate their theoretical findings.

**Strengths:**

- The paper is well-written overall and the proofs are well-structured and easy to follow.
 - The authors show that even with unbounded model dimension, single-head attention-only transformers are not able to learn the ESP task for cyclic graphs (but they are able to easily do so for DAGs).
 - The authors also show that the same model with two attention heads is able to learn the ESP task on cyclic graphs with limited model dimension and precision.
 - Experiments are conducted to corroborate their theoretical observations.

**Weaknesses:**

- The scope and implications of the contributions are not immediately obvious (or they seem to be somewhat limited). Perhaps some discussion about how the findings would generalize to non-attention-only transformers or transformers with more than one layer would help.
 - It is unclear why it is not possible to write a construction that is agnostic to the distribution of the graphs (see below).

**Questions:**

In the ESP task, all of the information needed to solve the problem is given in the model's input: the ordered pair representing the edge as well as the selector. Then why is the model's ability to solve this task dependent on the distribution of edges/graph structure? Is it not possible to write/learn a distribution-agnostic circuit to solve ESP in one layer? For example, if the query token were excluded from the input (i.e., the input only consists of the vertex pair and the indicator), it would be straightforward to construct a single attention-head transformer to solve the task in a distribution-agnostic fashion: Set the token embedding for vertex v_i to be the unit vector with 1 in dimension i; the token embedding for 1 and 2 are unit vectors with 1 in dimensions n and n+1, respectively; the position embedding for position i is a unit vector with 1 in dimension n+2+i, and so the embedding dimension is n+5. The attention matrix A is set as follows: A[n+0,n+2+0]=C, A[n+1,n+2+1]=C, for some positive constant C and zeros elsewhere. The last token (the selector) would simply copy from the token at position 0 or 1 depending on the value of the selector. Given such a construction exists, why is it worthwhile to consider the case where the last token must be a query token?

While the work focuses on attention-only transformers, since arbitrarily wide FF layers are universal approximators, it would still be interesting to explore how the analysis would change if we allow for bounded width FF layers. Would the ESP problem become much easier to solve? Similarly, would an attention-only transformer with one attention head and two layers be able to solve the ESP problem on cyclic graphs?

The font size in Figure 1 is rather small which makes it somewhat difficult to read (especially the expression in grey).

Figure 2: The font size for the percentages is a bit too small.

---

> ### Author Response · Authors · 2025-11-19
> **Response to WJsn**
>
> We thank you for the thoughtful evaluation, the positive comments on the clarity of our writing and proofs, and the insightful questions regarding our results. We address your points below and hope these clarifications resolve the concerns raised and allow you to consider increasing your support for the paper.
>
>
> **Q1.** How do the results generalize beyond attention-only transformers?
>
> > **A1.** We agree that extending our results to deeper or more general transformer architectures would further strengthen the work. However, the attention-only setting has been a standard analytical choice in prior theoretical studies (cited in the paper), precisely because it isolates the core computational properties of multi-head attention without the confounding effects of universal-approximator FFNs. Our lower bound (Theorem 2) is *unconditional*: it holds for arbitrarily large width and precision, indicating that the limitation we uncover is a structural property of attention itself rather than an artifact of model capacity. As noted in our official comment, the same geometric intuition explaining the failure of 1-head models on cyclic graphs may extend to higher-order structures, providing a pathway to generalizing our impossibility results to deeper or more complex transformer architectures.
>
> **Q2.** Why ESP? Why is it an interesting problem to study?
>
> **Q2.1.** Why fix the query token?
>
> > **A2.1.** In addition to the motivation we explain in the meta-response, this design choice also emerged from our empirical observations while studying whether single-head transformer models can reliably reverse (and copy) variable-length random paths extracted from a large bidirected graph. We found that the querying token at each time step behaves essentially as an **arbitrary** vertex of the graph. Consider the following three outputs from a transformer-model:
> >
> >     s··q··f··d··k··c··b··%··b··c··k··d··f··q··->··[s]··(token at position 1)
> >     ^
> >     r··w··h··k··a··q··d··f··i··%··i··f··d··q··->··[a]··(token at position 5)
> >     ············^
> >     f··e··k··s··n··b··o··h··g··q··l··%··l··q··->··[g]··(token at position 9)
> >     ························^
> >
> > Here `%` denotes a delimiter marking the end of a path. Across these three outputs, a **single-head** transformer-model uses **the same** querying vector ($\text{embed}(q)\ * W_Q$) to "copy" from three different positions. The correct prediction therefore cannot be attributed to any meaningful key–query interaction with the querying token. When the graph is large, the query token is effectively a random symbol from the vertex set. In other words: **the query token contributes very little information toward copying from the correct position.**
> >
> > Fixing the query token to a designated symbol \( \# \) therefore standardizes the query vector and allows us to cleanly isolate the core limitations of the attention mechanism.
>
> **Q2.2.** Why study graphs?
>
> > **A2.2.** We use directed graphs as our domain because they are a fundamental abstraction for **structured reasoning**. Our task, the **Endpoint Selection Problem (ESP)**, is a simple "test case" that probes a model's ability to perform graph traversal and selection primitives which serve as the building blocks for many higher-level reasoning tasks as noted in our paper.
> >
> > Our analysis reveals that the model's limitations are tied to the underlying graph's structure for two main reasons:
> >
> > - **NP-Completeness of even approximating optimal 1-head models:**
> > Please refer to the meta-response for more details.
> >
> > - **A Hierarchy of Capabilities:** Our core impossibility result provides a framework for extending our research. Just as cycles represent a fundamental structural barrier for 1-head models, we suggest that there are analogous, more complex "higher-order" graph structures that will extend impossibility result for $k$-heads. Identifying these boundaries is key to establishing a **fine-grained hierarchy of transformer architectures**, allowing us to characterize how each additional head expands the class of solvable problems.
>
> **Q3.** Why No FFN?
>
> > **A3.** Our focus in this work is on unbounded-width models and attention mechanisms.  So, we removed FFN confounds since unbounded-width FFNs are universal approximators.  Your suggestion to consider bounded-width FFNs is well-taken.  Indeed, we have conducted preliminary experiments to test whether adding a 2-layer FFN helps a single-head transformer to solve ESP. Across multiple settings—ranging from 130K-parameter to 8M-parameter MLPs, we have found that FFNs only help when they are extremely wide with respect to the number of nodes of the Graph used for ESP.  These experiments and accompanying theoretical analyses are directions for future research.  (Re your question about 2-layer attention-only models, we believe a 2-layer model can solve ESP on cyclic graphs, though it may struggle on higher-order generalizations of ESP.)
>
> **Q4.** Font size?
>
> > **A3.** Fixed

---

> ### Comment · Reviewer_WJsn · 2025-11-25
>
> I appreciate the author's thoughtful response to the questions in my review. Overall, I still feel that the motivation for the analysis is lacking. Are there any more tangible/realistic implications of the findings of the analysis?
>
> For example, the authors state:
> > As noted in our official comment, the same geometric intuition explaining the failure of 1-head models on cyclic graphs may extend to higher-order structures, providing a pathway to generalizing our impossibility results to deeper or more complex transformer architectures.
>
> But it is unclear to me what these generalized results look like. Is there an example of a more impactful/realistic task for which the proposed method would be able to derive a similar bound?
>
> ### Why fix the query token?
>
> The authors cite the reversal task as motivation for the use of a query token. However, the reversal task is such that for a given sequence of input tokens, followed by a separator (e.g., "%"), the model must output the reversed list of tokens. Adding a query token to every forward pass of the model would alter the task and deviate from the spirit of the task. Therefore, this is not very compelling motivation for the addition of a query token. The response does not really answer the question: Given the fact that ESP can be solved with a single attention layer without a query token, why is it interesting/worthwhile to consider the case where the query token is included? (especially since such a token does not typically appear in other canonical tasks such as reversal)
>
> ### The effect of bounded-width FFN
>
> I also appreciate the additional experiments on transformers with bounded-width FFNs. However, detailed results from these experiments are missing, and the authors did not indicate that these results would be added to the paper's revision.

---

> > ### Author Response · Authors · 2025-12-02
> > **Followup Response to Reviewer WJsn**
> >
> > We thank the reviewer for their continuing feedback, and for engaging with us to seek further clarifications.  We address all of the questions below.
> >
> > **Q1.** *Are there any more tangible/realistic implications of the findings of the analysis? Is there an example of a more impactful/realistic task for which the proposed method would be able to derive a similar bound?*
> >
> > **A1.** Yes.  In fact, the revision of our paper that we uploaded in the rebuttal includes a proof that the well-studied 2-hop induction head problem cannot be solved by a 1-head 1-layer attention-only model, even for unbounded dimension (width) and unbounded precision.  The original submission had mentioned a reduction to this effect; we have clarified and highlighted this in the revision.  See Corollary 1, with full proof in Appendix A.  Note that the 2-hop induction head problem definition is identical to as specified in previous work **(Olsson et al., 2022)** and extensively studied in the multi-hop induction framework **(Sanford, Hsu, and Telgarsky, 2024d)**.  (We emphasize that no auxiliary query token is included in the input.)
> >
> > In addition, we are exploring the following directions in our ongoing research.  Though we have made progress, these are not advanced enough to merit inclusion in a revision; we also feel including them will significantly change the scope of the paper.
> >
> > - We have derived the conditions on the embedding vectors obtained by a k-head model to solve "direct product" versions of ESP.  These conditions are generalizations of the inequalities derived in the proof of Theorem 2 in the paper.
> > - We have empirical results for list reversals with different length lists (with the standard definition using a separator token between a list and its reversal, thus not including a query token)
> > - We also have empirical results for identifying $s$-$t$ paths and their reversals in directed acyclic graphs; these problems are variations of the list reversal problem.
> >
> > **Q2.** *Query token and the reversal task, and motivation for the query token.*
> >
> > **A2.**
> >
> > - First, we want to highlight a possible misunderstanding.  As you mention, the reversal task consists of a sequence of input tokens, followed by a separator (e.g., "%").  Indeed, *this separator plays the role of the query token*.  There is no need to add an "additional" query token, so *the definition of the task remains the same*, exactly as you intended it to be.
> > - Though the query token is integral to the ESP definition, our geometric framework for analyzing the models is orthogonal, broad and amenable to generalization in different ways.  Indeed, as we mention above, our paper includes a proof (Corollary 1 and Appendix A) for the well-studied 2-hop induction head problem (which does not have a dedicated query token).
> > - Our motivation for defining ESP (with query token) is to obtain what is arguably the simplest problem yielding the impossibility and intractability results with attention-only models that we have derived. The query token elicits the weakness of the single attention-layer architecture.  In our ongoing work, we are probing more elaborate transformer architectures through hypergraph variants (via direct products), multi-way selection (multiple indexes) and multi-hop versions (similar to 𝑘-hop induction heads).
> > - Finally, many inputs of transformer models often have an "eos" token or a final instruction string similar to a query token.
> >
> >
> > **Q3.** *I also appreciate the additional experiments on transformers with bounded-width FFNs. However, detailed results from these experiments are missing, and the authors did not indicate that these results would be added to the paper's revision.*
> >
> > **A3.** Yes, we have added experimental results for a transformer model with bounded-width FFNs.  In particular, we have included a plot in Appendix C.2, which indicates that ESP can be solved by 1-head transformers with FFNs with the number of parameters being comparable to that of transformers with two attention heads.

---

### Author Response · Authors · 2025-11-19
**Meta-response**

We would like to thank all reviewers for their thorough reading and assessment of our paper and for the constructive feedback they have provided. We are encouraged that reviewers consistently found the work to be technically sound and relevant to the study of transformer-models, as reflected in the “soundness” and “contribution” scores. We also appreciate the positive comments on the clarity of our writing, highlighted in the strengths sections of multiple reviewers, and the recognition of our combination of rigorous theoretical analysis with empirical evaluation, a theme noted across all four reviews.

We have uploaded a new revision, which addresses the suggestions and concerns of the reviewiers.  All of our additions to the paper are marked in red.

We respond to each reviewer’s questions individually in separate official responses.  In this official comment, we would like to highlight three themes that appeared across multiple reviews and that we address more directly below.

### 1. On ESP's Query Token and Graph-Theoretic Nature

We originally defined ESP for three reasons:
* **Hardness**: ESP (with query token) is arguably the simplest problem yielding the impossibility and intractability results with attention-only models that we have derived. As the reviewers correctly observe, ESP instances redefined without the query token are solvable by attending to the position embedding.
* **Reductions**: Fixing the query token in our ESP formulation, critically, enables us to extend our lower bounds to related problems that lack explicit query tokens, in particular, the much-studied 2-hop induction head problem (Corollary 1).
* **Hierarchies**: Our formulation naturally generalizes to probe more elaborate transformer architectures through hypergraph variants (via direct products), multi-way selection (multiple indexes) and multi-hop versions (similar to 𝑘-hop induction heads).

In the course of studying ESP a deeper connection to graph theory emerged: the optimal 1-head model's error is *exactly* (1/2 - |MAS|/(2m)), where MAS is the Maximum Acyclic Subgraph and m the number of arcs, an NP-complete problem (Theorem 1 and Corollary 3). This isn't merely a task defined over graphs; the transformer's representational capacity is *intrinsically* tied to the graph's structural properties. The model cannot achieve distribution-agnostic performance precisely because cycles create geometric constraints in the attention mechanism's representational space (Lemma 2). Just as cycles represent a fundamental structural barrier for 1-head models, we suggest that there are analogous, more complex "higher-order" graph structures that will prove impossible for 2-head or $k$-head models to solve. Identifying these boundaries is key to establishing a **fine-grained hierarchy of transformer architectures**, potentially allowing us to characterize how each additional head acts as a discrete unit of computational power that expands the class of solvable problems.

### 2. On Attention-Only Models and Practical Relevance

We acknowledge the limitation but argue this restriction is methodologically valuable:
- **Theoretical tractability:** Even this simplified model is challenging to analyze rigorously. Our unconditional impossibility result (Theorem 2) required novel geometric techniques that may extend to richer architectures.
- **Isolating attention's role:** By removing FFN confounds, we precisely characterize what multi-head attention alone contributes, which is essential for understanding transformer mechanisms.
- **Foundation for hierarchies:** We conjecture our techniques extend to establish a strict k-head hierarchy, potentially revealing fundamental scaling laws in transformer expressivity.

### 3. On Related Work and Contribution Positioning

We thank the reviewers for highlighting [1]. While their work on Index uses more realistic models, our contributions remain valuable because:
- Our results are **unconditional**: they hold for unbounded dimension and precision, unlike prior conditional bounds
- We establish a direct connection between transformer expressivity and computational complexity (**NP-completeness**)
- ESP, though superficially similar to Index, exhibits fundamentally different behavior tied to **graph-theoretic** properties


### Summary

Our work reveals that even minimal architectural choices can fundamentally alter a transformer's computational boundaries, with precise connections to classical complexity theory. This provides a new lens for understanding transformer scaling and design choices.

[1] Bhattamishra, S., Hahn, M., Blunsom, P., & Kanade, V. (2024). Separations in the representational capabilities of transformers and recurrent architectures. Advances in Neural Information Processing Systems, 37, 36002-36045.

---

### Author Response · Authors · 2025-12-02
**Comment to Area Chair**

Thank you for your time and attention to our paper, the reviews, and the rebuttal discussions.  We recognize that Area Chairs are overloaded and are being asked to make decisions under fraught conditions.  We really appreciate your service.

We are also grateful to the reviewers for their thorough reviews, their insightful feedback, and the opportunity to engage in discussion with them during the rebuttal process.

We are confident that our rebuttal responses and the revised paper have addressed all of the key comments of the reviewers, and we hope you can gather this sense from your review of the transcripts.  We refer you to the paper (all post-rebuttal revisions marked in red) and the metaresponse comment for the significance of our contributions.  In this comment, we highlight a few important points about the discussions with the reviewers.

* **Ratings:** The original ratings of the paper were 4, 6, 4, 4.  Subsequent to our detailed rebuttal responses (a metaresponse and 5 individual author comments), the ratings were 4, 6, 4, 8.

We kindly request the Area Chair to give weight to the feedback and final discussion with Reviewers 1 and 4. We believe that Reviewers 2 (already positive) and 3 had not yet had a chance to engage. The discussion phase revealed a significant positive trajectory for the paper, evidenced by: (a) Reviewer 4 increasing their score from 4 to 8 after reading the rebuttal and engaging in subsequent discussion; (b) Reviewer 1 expressing positiveness and posing follow-up questions, indicating active engagement post-rebuttal.

* **Reviewer 4:** Reviewer kCWe wrote in their original evaluation, "*I actually think this is a nice paper, but the relationship to [1] really needs to be addressed before publication, hence my initial low score. I am open to recommending acceptance if the authors can address these concerns in rebuttal/discussion and amend the paper accordingly.*"  We gave a detailed comparison in the rebuttal and also suitably revised the paper, which convinced reviewer kCWe to respond "*Given the updates, I'm happy to increase my score*" and increase their score.
* **Reviewer 1**: Reviewer WJsn sought further clarification in a comment.  We have submitted a new author comment, which we believe fully addresses their concerns. In particular, one of their comments (about additional problems where our geometric technique could apply) was already addressed in the revision we had posted with our rebuttal, and another of their comments (about query tokens in list reversal) mis-stated our point in the discussion, when in fact both WJsn and we are in full agreement about the input for list reversals.  Our response to Reviewer WJsn clarifies how all their points are addressed.

We trust you will take the full picture into account, including the revised ratings and the rebuttal exchanges, and give our paper the fair shot that it deserves in the decision process.

---

### Meta-Review · Area_Chair_4Tf4 · 2026-01-06

**Summary:**

The paper presents a theoretical analysis of single-layer, attention-only transformers, establishing unconditional impossibility results for the Exact Selector Problem (ESP) and linking the model’s representational capacity to graph-theoretic properties . The decision process was heavily influenced by the tension between the paper’s rigorous theoretical depth, and the scope of the experimental setting. While initial concerns focused on the restrictiveness of the attention-only assumption and the specific formulation of the ESP task, the rebuttal successfully clarified the distinction from prior work and the theoretical necessity of the chosen constraints. Notably, the rebuttal led to a significant positive re-evaluation, though questions regarding the generalization to more complex, practical architectures remain a point of discussion.

**Reviewer Concerns:**

Artificiality of the Problem Setup: Reviewer 1 noted that the difficulty of the ESP task is highly sensitive to the exact formulation. While the authors admit this is intentional to isolate specific mechanisms, the concern persists that the derived "impossibility" results are heavily dependent on this specific, arguably artificial, constraint rather than being a general property of the architecture under broader conditions.

Connection to Graph Theory: The authors successfully expanded on the "distribution dependency" concern by demonstrating that the limitation is not merely distributional but structural—tied to the NP-completeness of the Maximum Acyclic Subgraph problem—thereby strengthening the theoretical significance of the work.

**Reviewer Scores:**

none

---

### Decision · Program_Chairs · 2026-01-26

Accept (Poster)